# REAL-WORLD DATA AND CALIBRATED SIMULATION SUITE FOR OFFLINE TRAINING OF REINFORCEMENT LEARNING AGENTS TO OPTIMIZE ENERGY AND EMISSION IN BUILDINGS FOR ENVIRONMENTAL SUSTAINABILITY

## ABSTRACT

Commercial office buildings contribute 17 percent of Carbon Emissions in the US, according to the US Energy Information Administration (EIA), and improving their efficiency will reduce their environmental burden and operating cost. A major contributor of energy consumption in these buildings are the Heating, Ventilation, and Air Conditioning (HVAC) devices. HVAC devices form a complex and interconnected thermodynamic system with the building and outside weather conditions, and current setpoint control policies are not fully optimized for minimizing energy use and carbon emission. Given a suitable training environment, a Reinforcement Learning (RL) agent is able to improve upon these policies, but training such a model, especially in a way that scales to thousands of buildings, presents many practical challenges. Most existing work on applying RL to this important task either makes use of proprietary data, or focuses on expensive and proprietary simulations that may not be grounded in the real world. We present the Smart Buildings Control Suite, the first open source interactive HVAC control dataset extracted from live sensor measurements of devices in real office buildings. The dataset consists of two components: six years of real-world historical data from three buildings, for offline RL, and a lightweight interactive simulator for each of these buildings, calibrated using the historical data, for online and model-based RL. For ease of use, our RL environments are all compatible with the OpenAI gym environment standard. We also demonstrate a novel method of calibrating the simulator, as well as baseline results on training an RL agent on the simulator, predicting real-world data, and training an RL agent directly from data. We believe this benchmark will accelerate progress and collaboration on building optimization and environmental sustainability research.

## 1 INTRODUCTION

Energy optimization and management in commercial buildings is a very important problem, whose importance is only growing with time. Buildings account for 37% of all US carbon emissions, with commercial buildings alone taking up a staggering 17% in 2023 (EIA). Reducing those emissions by even a small percentage can have a significant effect. In climates that are either very hot or very cold, energy consumption is much higher, and there is even more room to have a major impact. We believe this problem is one of the most important avenues for climate sustainability research, where even a small improvement over baseline policies can drastically reduce our carbon footprint.

In particular, HVAC systems account for 40-60% of energy use in buildings (Pérez-Lombard et al., 2008), and roughly 15% of the world's total energy consumption (Asim et al., 2022). Most office buildings are equipped with advanced HVAC devices, like Variable Air Volume (VAV) devices, Hot Water Systems, Air Conditioners and Air Handlers that are configured and tuned by the engineers, manufacturers, installers, and operators to run efficiently with the device's local control loops (McQuiston et al., 2023). However, integrating multiple HVAC devices from diverse vendors into a

building "system" requires technicians to program fixed operating conditions for these units, which may not be optimal for every building and every potential weather condition. Existing setpoint control policies are not optimal under all conditions, and the possibility exists that a machine learning model may be trained to continuously tune a small number of setpoints to achieve greater energy efficiency and reduced carbon emission.

Optimizing HVAC control has been an active research area for decades, and yet while AI has begun to revolutionize many industries, to date almost all HVAC systems remain the same as they were 30 years ago: despite all the literature on the topic, there is not a single solution that has been widely adopted in the real world.

One of the most significant factors limiting progress is the lack of a reliable public benchmark to test solutions against. Current work generally makes use of proprietary data and expensive (often also proprietary) simulations. This limits participation to those with exclusive access, and makes most claims difficult to verify and compare. A strong public dataset would facilitate collaborations between institutions, standardize research efforts, and allow for wider participation. Historically, much of progress in AI has been driven by easily accessible public benchmarks, from the ImageNet Challenge in Vision (Russakovsky et al., 2015), to the Atari57 suite in RL (Badia et al., 2020), and the GLUE Benchmark in language (Wang et al., 2018). A similar benchmark in HVAC control may help accelerate progress and finally lead to adoption of solutions in the real world.

We present The Smart Buildings Control Suite, a high quality, fully accessible, building control benchmark. The benchmark consists of two components:

- Real-world historical HVAC data, collected from three buildings over a six year period.

- A highly customizable and scalable HVAC and building simulator, with configurations corresponding to each of the above buildings

Our contributions include one of the first public real-world HVAC datasets, a highly customizable and scalable HVAC and building simulator, a rapid configuration method to customize the simulator to a particular building, a calibration method to improve this fidelity using real-world data, and an evaluation method to measure the simulator fidelity. The dataset contains information from three buildings in California, the largest of which is three stories and 118,086 ft$^2$. Using data we obtained from each building, we calibrate our simulator, and demonstrate using our evaluation pipeline that this significantly improves its fidelity to the real building. We provide pre-calibrated simulators for all of our buildings, as well as code to both reproduce the calibration procedure, and to calibrate the simulator to new scenarios. While our suite focuses on three buildings, our simulator is easily adaptable, allowing for the development of general purpose solutions that can be applied to any building. All the data and simulator code is open source and compatible with the OpenAI gym environment standard(Brockman et al., 2016), and data is available on the popular TensorFlow Datasets platform (TFDS) under the Creative Commons License.

We first give an overview of the problem and related work, and then present the structure of the data. Next we introduce the simulator, and discuss our configuration, calibration, and evaluation techniques. After that, we run through an example of the process of calibrating the simulator to real data, and finally we demonstrate success on three key benchmark tasks: training an RL agent on the calibrated simulator environment using Soft Actor Critic(Haarnoja et al., 2018), training a regression model to predict the real world dynamics, and training a Soft Actor Critic agent from the real world data via the regression model.

## 2   OPTIMIZING ENERGY AND EMISSION IN OFFICE BUILDINGS WITH RL

In this section we frame energy optimization in office buildings as an RL problem. We define the state of the office building $S_t$ at time $t$ as a fixed length vector of measurements from sensors on the building's devices, such as a specific VAV's zone air temperature, gas meter's flow rate, etc. The action on the building $A_t$ is a fixed-length vector of device setpoints selected by the agent at time $t$, such as the boiler supply water temperature setpoint, etc.

More generally, RL is a branch of machine learning that attempts to train an agent to choose the best actions to maximize some long-term, cumulative reward (Sutton & Barto, 2018). The agent observes

the state $S_t$ from the environment at time $t$, then chooses action $A_t$. The environment responds by transitioning to the next state $S_{t+1}$ and returns a reward (or penalty) after the action, $R_{t+1}$. Over time, the agent will explore the action space and learn to maximize the reward over the long term for each given state. A discount factor $\gamma$ reduces the value of future rewards amplifying the value of the near-term reward. When this cycle is repeated over multiple episodes, the agent converges on a state-action policy that maximizes the long-term reward.

This sequence is often formalized as the Markov Decision Process (MDP) (Garcia & Rachelson, 2013), described by the tuple $(S, A, p, R)$ where the state space is continuous (e.g., temperatures, flow rates, etc.) and the action space is continuous (e.g., setpoint temperatures) and the transition probability $p : S \times S \times A \to [0, 1]$ represents the probability density of the next state $S_{t+1}$ from taking action $A_t$ on the current state $S_t$. The reward function $R : S \times A \to [R_{min}, R_{max}]$ emits a single scalar value at each time $t$. The agent is acting under a policy $\pi_\theta(A_t|S_t)$ parameterized by $\theta$ that represents the probability of taking action $A_t$ from state $S_t$. The goal of an RL agent is to find the policy that maximizes the expected long-term cumulative, discounted reward. The set of parameters $\boldsymbol{\theta}^*$ of the optimal policy can be expressed as:

$$\boldsymbol{\theta}^* = \arg\max_\theta \mathbb{E}_{\tau \sim \pi_\theta(\tau)} \left[ \sum_t \gamma^t R(S_t, A_t) \right]$$

where $\theta$ is the current policy parameter, and $\tau$ is a trajectory of states, actions, and rewards over sequential time steps $t$. In order to converge to the optimal policy, the agent requires many training iterations to explore the policy space, making online training directly on the real-world building from scratch inefficient, dangerous, impracticable, and likely impossible. Therefore, it is necessary to enable offline learning, where the agent can train in an efficient sandbox environment that adequately emulates the dynamics of the building before being deployed to the real world.

**Reward Function** RL generally requires a single scalar reward signal, $R_t(S_t, A_t)$ that indicates the quality of taking action $A_t$ in state $S_t$. We thus define a custom feedback signal, $R_{3C}$, as a weighted sum of negative cost functions for carbon emission, energy cost, and comfort levels within the building, which we dubbed the 3C Reward. It is governed by the following equation:

$$R_{3C} = u \times C_1 + v \times C_2 + w \times C_3$$

where $C_1$ represents normalized comfort conditions, $C_2$ normalized energy cost and $C_3$ normalized carbon emission. Constants $u$, $v$, $w$ represent operator preferences, allowing them to weight the relative importance of cost, comfort and carbon consumption. $R_{3C} = 0$ when no energy is consumed, no carbon is emitted, and all occupied zones are in setpoint bounds, and negative otherwise. For more details, and equations governing how we normalize and measure these quantities, see Appendix A.

## 3 RELATED WORKS

Considerable attention has been paid to HVAC control (Fong et al., 2006) in recent years (Kim et al., 2022), and while alternative approaches exist, such as model predictive control (Taheri et al., 2022), a growing portion of the literature has considered how RL and its various associated algorithms can be leveraged (Yu et al., 2021; Mason & Grijalva, 2019; Yu et al., 2020; Gao & Wang, 2023; Wang et al., 2023; Vázquez-Canteli & Nagy, 2019; Zhang et al., 2019b; Fang et al., 2022; Zhang et al., 2019b). As mentioned above, a central requirement in RL is the offline environment that trains the RL agent. Several methods have been proposed, largely falling under three broad categories.

**Data-driven Emulators** Some works attempt to learn a dynamics as a multivariate regression model from real-world data (Zou et al., 2020; Zhang et al., 2019a), often using recurrent neural network architecture, such as Long Short-Term Memory (LSTM) (Velswamy et al., 2017; Sendra-Arranz & Gutiérrez, 2020; Zhuang et al., 2023). The difficulty here is that data-driven models often do not generalize well to circumstances outside the training distribution, especially since they are not physics based.

**Offline RL** The second approach is to train the agent directly from the historical real-world data, without ever producing an interactive environment (Chen et al., 2020; 2023; Blad et al., 2022). While the real-world data is obviously of high accuracy and quality, this presents a major challenge, since the agent cannot take actions in the real world and interact with any form of an environment. This inability to explore severely limits its ability to improve over the baseline policy producing the real-world data (Levine et al., 2020). Furthermore, prior to our work, there are few public datasets available.

**Physics-based Simulation** HVAC system simulation has long been studied (Trčka & Hensen, 2010; Riederer, 2005; Park et al., 1985; Trčka et al., 2009; Husaunndee et al., 1997; Trcka et al., 2007; Blonsky et al., 2021). EnergyPlus (Crawley et al., 2001), a high-fidelity simulator developed by the Department of Energy, is commonly used (Wei et al., 2017; Azuatalam et al., 2020; Zhao et al., 2015; Wani et al., 2019; Basarkar, 2011), but suffers from scalability and configuration challenges.

To overcome the limitations of each of the above three methods, some work has proposed a hybrid approach (Zhao et al., 2021; Balali et al., 2023; Goldfeder & Sipple, 2023; Zhang et al., 2023; Klanatsky et al., 2023; Drgoňa et al., 2021), and indeed this is the category our work falls under. What is unique about our approach is the use of a physics based simulator that achieves an ideal balance between speed of configuration, and fidelity to the real world. Our simulator is lightweight enough to be configured to an arbitrary building in a matter of hours, and using our calibration process based on real-world data, accurate enough to train an effective control agent off-line. This allows our solution to be highly scalable, like the first two approaches, but still rooted in physics, and demonstrably calibrated, like the third approach.

Various works have also discussed how exactly to apply RL to an HVAC environment, such as what sort of agent to train. Inspired by prior effective use of Soft Actor Critic (SAC) on related problems (Kathirgamanathan et al., 2021; Coraci et al., 2021; Campos et al., 2022; Biemann et al., 2021), we chose to demo our environment using a SAC agent.

**Prior Datasets** While many building datasets exist (Ye et al., 2019), most either have a different focus (Sachs et al., 2012; Urban et al.; Kriechbaumer & Jacobsen, 2018; Granderson et al., 2023), do not contain sufficient HVAC information (Miller et al., 2020; Mathew et al., 2015; Rashid et al., 2019; Jazizadeh et al., 2018; Sartori et al., 2023), are focused on residential buildings (Murray et al., 2017; Barker et al., 2012; Meinrenken et al., 2020) or non-standard buildings (Pettit et al., 2014; Naug & Chandan), or are simulated (Field et al., 2010; Bakker et al., 2022). Even the few datasets directly relevant (Luo et al., 2022; Heer et al., 2024) are non-interactive. As far as we are aware, we present the first HVAC control benchmark that has high quality real-world data with computationally cheap simulations of the same buildings, allowing for both real-world grounding and interactive control experiments.

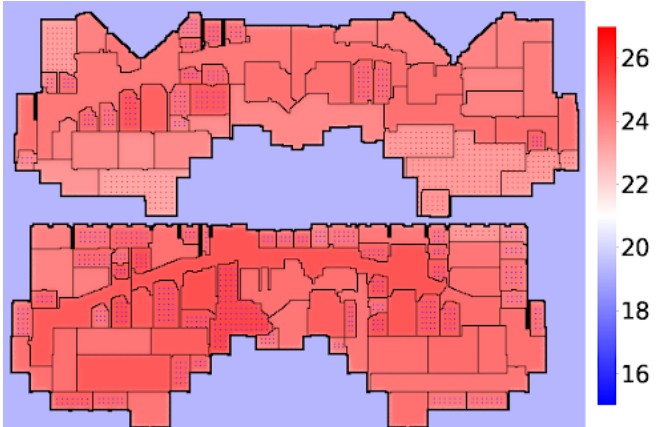

Figure 1: Example Visualization of an Environment. Blue represents colder temperatures, red warmer. Blue and red dots inside the building indicate diffusers that are dispensing cold and warm air respectively.

## 4 THE DATASET STRUCTURE

Both the real-world data and simulated data are given in the same format. Following the RL paradigm, data is provided as a series of observations, actions, and rewards. In the case of the

real-world data, this comes in the form of static historical episodes, where the actions follow the baseline policy in the building, and in the case of the simulator, as a proper interactive RL environment where actions can be taken in real time.

To make the task as realistic as possible, we formatted the data to closely resemble the real-world building API, so that a user can mimic interacting with the building. All of our data is formatted to be compliant with the popular open source Google Digital Buildings Ontology (DBO). The agent communicates with the building using the Protobuf open source serialization format(Google). The agent can send information requests to the building, asking for structural information, such as the number of devices, and telemetry information, such as the value of a particular sensor, and the building sends back a response, containing the requested information. The agent can also request that a setpoint be changed to a new value, and the building will respond if the change was successful.

Following the RL paradigm, the data in our dataset falls under the following categories:

1. **Environment Data** or each building environment, the dataset contains information on all HVAC zones and HVAC devices. For zones this includes the name and size of each zone, as well has how many devices are contained within it. For devices, this includes the zone the device is associated with, as well as every device sensor and setpoint.

2. **Observation Data** Observations consist of the measurements from all devices in the building (VAV's zone air temperature, gas meter's flow rate, etc.), provided at each time step.

3. **Action Data** The device setpoint values that the agent wants to set, provided at each timestep

4. **Reward Data** Information used to calculate the reward, as expressed in cost in dollars, carbon footprint, and comfort level of occupants, provided at each time step

The dataset currently consists of six years of data from three buildings. The details are in Table 1. For more details regarding the format of the data, including definitions and examples of each type of proto, see appendix B.

**Data Visualization** We also present a data visualization module, compatible both for viewing the real-world historical data, as well as visualizing the state of the simulator, as shown in Figure 1. Given an observation of a building environment, our visualization module renders a two dimensional heat-map view of the building. This greatly aids in understanding the data, and is invaluable in understanding how a particular policy is behaving.

Table 1: Building Information

| BUILDING | FT$^2$ | FLOORS | DEVICES |
|---|---|---|---|
| SB1 | 93,858 | 2 | 170 |
| SB2 | 62,613 | 1 | 152 |
| SB3 | 118,086 | 3 | 152 |

## 5 SIMULATOR DESIGN CONSIDERATIONS

A fundamental trade-off when designing a simulator is speed versus fidelity, as depicted in Figure 2. Fidelity is the simulator's ability to reproduce the building's true dynamics that affect the optimization process. Speed refers to both simulator configuration time, i.e., the time required to configure a simulator for a target building, and the agent training time, i.e., the time necessary for the agent to optimize its policy using the simulator.

Every building is unique, due to its physical layout, equipment, and location. Fully customizing a high fidelity simulation to a specific target building requires nearly exhaustive knowledge of the building structure, materials, location, etc., some of which are unknowable, especially for legacy office buildings. This requires manual "guesstimation", which can erode the accuracy promised by high-fidelity simulation. In general, the configuration time required for high-fidelity simulations limits their utility for deploying RL-based optimization to many buildings. High-fidelity simulations also are affected by computational demand and long execution times.

Alternatively, we propose a fast, low-to-medium-fidelity simulation model that was useful in addressing various design decisions, such as the reward function, the modeling of different algorithms. and for end-to-end testing. The simulation is built on a 2D finite-difference (FD) grid that models

thermal diffusion, and a simplified HVAC model that generates or removes heat on special "diffuser" control volumes (CV) in the FD grid. For more details on design considerations, see Appendix C.

While the uncalibrated simulator is of low-to-medium fidelity, the key additional factor is data. We collect recorded observations from the target building under baseline control, and use that data to **calibrate** the simulator, by adjusting the simulator's physical parameters to minimize difference between real and simulated data. We believe this approach hits the sweet spot in this tradeoff, enabling scalability, while maintaining a high enough level of fidelity to train an improved policy.

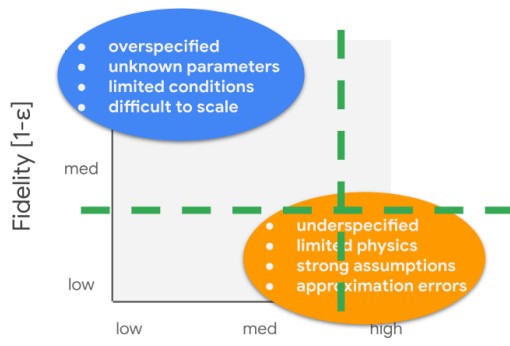

Figure 2: Simulation Fidelity vs. Execution Speed. The ideal operating point for training RL agents for energy and emission efficiency is a tradeoff between fidelity, depicted as 1 minus a normalized error $\epsilon$ between simulation and real, and execution speed, as measured by the number of training steps per second. Additional consideration also includes the time to configure a custom simulator for the target building. While many approaches tend to favor high-fidelity over execution, speed, our approach argues a low-to-medium fidelity that has a medium-to-high speed is most suitable for training an RL agent.

## 6 A Lightweight, Calibrated Simulation

Our goal is to develop a method for applying RL at scale to commercial buildings. To this end, we put forth the following requirements for this to be feasible: We must have an easily customizable simulated environment to train the agent, with high enough fidelity to train an improved control agent. To meet these desiderata, we designed a light weight simulator based on finite differences approximation of heat exchange, building upon earlier work (Goldfeder & Sipple, 2023). We proposed a simple automated procedure to go from building floor plans to a custom simulator in a short time, and we designed a calibration and evaluation pipeline, to use data to fine tune the simulation to better match the real world. What follows is a description of our implementation.

**Thermal Model for the Simulation** As a template for developing simulators that represent target buildings, we start with a general-purpose high-level thermal model for simulating office buildings, illustrated in Figure 3. In this thermal cycle, we highlight significant energy consumers as follows. The boiler burns natural gas to heat the water, $\dot{Q}_b$ . Water pumps consume electricity $\dot{W}_{b,p}$ to circulate heating water through the VAVs. The air handler fans consume electricity $\dot{W}_{b,in}$ , $\dot{W}_{b,out}$ to circulate the air through the VAVs. A motor drives the chiller's compressor to operate a refrigeration cycle, consuming electricity $\dot{W}_c$ . In some buildings coolant is circulated through the air handlers with pumps that consume electricity, $\dot{W}_{c,p}$.

We selected **water supply temperature** $\hat{T}_b$ and the **air handler supply temperature** $\hat{T}_s$ as agent actions because they affect the balance of electricity and natural gas consumption, they affect multiple device interactions, and they affect occupant comfort. Greater efficiencies can be achieved with these setpoints by choosing the ideal times and values to warm up and cool down the building in the workday mornings and evenings. Further tradeoffs include balancing the thermal load between hot water heating with natural gas and supply air heating with electricity using the air conditioner or heat pump units.

**Finite Differences Approximation** The diffusion of thermal energy in time and space of the building can be approximated using the method of Finite Differences (FD)(Sparrow, 1993; Lomax et al., 2002), and applying an energy balance. This method divides each floor of the building into a grid of three-dimensional control volumes and applies thermal diffusion equations to estimate the temperature of each control volume. By assuming each floor is adiabatically isolated, (i.e., no heat is transferred between floors), we can simplify the three-spatial dimensions into a spatial two-dimensional heat transfer problem. Each control volume is a narrow volume bounded horizontally, parameter-

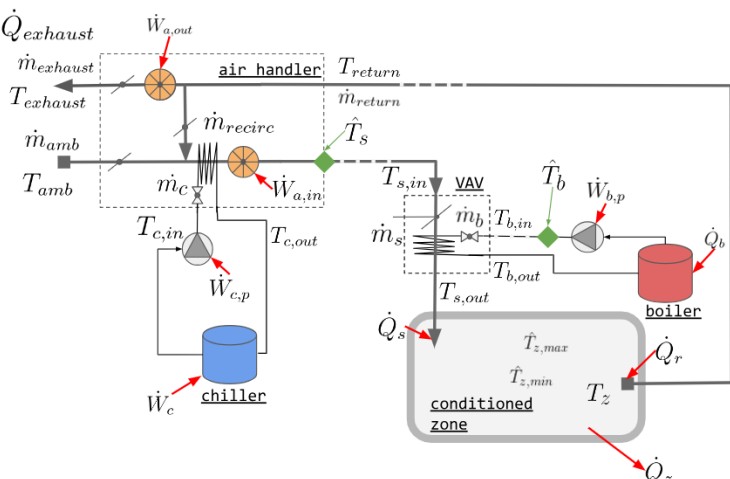

Figure 3: Thermal model for simulation. A building consists of conditioned zones, where the mean temperature of the zone $T_z$ should be within upper and lower setpoints, $\hat{T}_{z,max}$ and $\hat{T}_{z,min}$. Thermal power for heating or cooling the room is supplied to each zone, $\dot{Q}_s$, and recirculated from the zone, $\dot{Q}_r$ from the HVAC system, with additional thermal exchange $\dot{Q}_z$ from walls, doors, etc. The Air Handler supplies the building with air at supply air temperature setpoint $\hat{T}_s$ drawing fresh air, $\dot{m}_{amb}$, at ambient temperatures, $T_{amb}$, and returning exhaust air $\dot{m}_{exhaust}$ at temperature $T_{exhaust}$ to the outside using intake and exhaust fans, $\dot{W}_{a,in}$ and $\dot{W}_{a,out}$. A fraction of the return air can be recirculated, $\dot{m}_{recirc}$. Central air conditioning is achieved with a chiller and pump that joins a refrigeration cycle to the supply air, consuming electrical energy for the AC compressor $\dot{W}_c$ and coolant circulation, $\dot{W}_{c,p}$. The hot water cycle consists of a boiler that maintains the supply water temperature at $T_b$ heated by natural gas power $\dot{Q}_b$, and a pump that circulates hot water through the building, with electrical power $\dot{W}_{b,p}$. Supply air is delivered to the zones through Variable Air Volume (VAV) devices.

ized by $\Delta x^2$, and vertically by the height of the floor. The energy balance, shown below, is applied to each discrete control volume in the FD grid, and consists of the following components: (a) the thermal exchange across each face of the four participating faces control volume via conduction or convection $Q_1$, $Q_2$, $Q_3$, $Q_4$, (b) the change in internal energy over time in the control volume $Mc\frac{\Delta T}{\Delta t}$, and (c) an external energy source that enables applying local thermal energy from the HVAC model only for those control volumes that include an airflow diffuser, $Q_{ext}$. The equation is $Q_{ext} + Q_1 + Q_2 + Q_3 + Q_4 = Mc\frac{\Delta T}{\Delta t}$, where $M$ is the mass and $c$ is the heat capacity of the control volume, $\Delta T$ is the temperature change from the prior timestep and $\Delta t$ is the timestep interval.

The thermal exchange in (a) is calculated using Fourier's law of steady conduction in the interior control volumes (walls and interior air), parameterized by the conductivity of the volume, and the exchange across the exterior faces of control volumes are calculated using the forced convection equation, parameterized by the convection coefficient, which approximates winds and currents surrounding the building. The change in internal energy (b) is parameterized by the density, and heat capacity of the control volume. Finally, the thermal energy associated with the VAV (c) is equally distributed to all associated control volumes that have a diffuser. Thermal diffusion within the building is mainly accomplished via forced or natural convection currents, which can be notoriously difficult to estimate accurately. We note that heat transfer using air circulation is effectively the exchange of air mass between control volumes, which we approximate by a randomized shuffling of air within thermal zones, parameterized by a shuffle probability and radius. For more details on this approximation and associated equations, see Appendix D.

**Simulator Configuration** For RL to scale to many buildings, it is critical to be able to easily and rapidly configure the simulator to any arbitrary building. We designed a procedure that, given floor-plans and HVAC layout information, enables generating a fully specified simulation very rapidly.

For example, on SB1, consisting of two floors and 170 devices, a single technician was able to configure the simulator in under three hours. Details of this procedure are provided in Appendix E.

**Simulator Calibration and Evaluation** In order to calibrate the simulator to the real world using data, we must have a metric with which to evaluate our simulator's fidelity, and an optimization method to improve our simulator on this metric.

$N$**-Step Evaluation** We propose a novel evaluation procedure, based on $N$-step prediction. Each iteration of our simulator was designed to represent a five-minute interval, and our real-world data is also obtained in five-minute intervals. To evaluate the simulator, we take a chunk of real data, consisting of $N$ consecutive observations. We then initialize the simulator so that its initial state matches that of the starting observation, and run the simulator for $N$ steps, replaying the same HVAC policy as was used in the real world. We then calculate our simulation fidelity metric, which is the mean absolute error of the temperatures in each temperature sensor at each time step, averaged over time. More formally, we define the Temporal Spatial Mean Absolute Error (TS-MAE) of $Z$ zones over $N$ timesteps as:

$$\epsilon = \sum_{t=1}^{N} \frac{1}{N} \left[ \frac{1}{Z} \sum_{z=1}^{Z} |T_{real,t,z} - T_{sim,t,z}| \right] \tag{1}$$

Where $T_{real,t,z}$ is the measured zone air temperature for zone $z$ at timestamp $t$, and $T_{sim,t,z} = \frac{1}{|C_z|} \sum_{c=1}^{C_z} T_{t,c}$ is the mean temperature of all control volumes $C_z$ in zone $z$ at time $t$.

**Hyperparameter Calibration** Once we defined our simulation fidelity metric, the TS-MAE, we can attempt to minimize this error, thus improving fidelity, by hyperparameter tuning several physical constants and other variables using black-box optimization methods. We chose the method outlined in Golovin et. al. (Golovin et al., 2017), which automatically chooses the most appropriate strategy from a variety of popular algorithms.

# 7    SIMULATOR CALIBRATION

We now provide a full end-to-end demonstration of our calibration procedure, and show that our simulator, when tuned and calibrated, is able to make useful real-world predictions, and can train an RL agent to produce an improved policy over the baseline.

**Setup** We calibrated the simulator using data from SB1, with two stories, a combined surface area of 93,858 square feet, and 170 HVAC devices. Using the configuration pipeline, we went from floor plan blueprints to a fully configured simulator for this building, a process that took a single technician less than three hours to complete.

**Calibration Data** To calibrate our simulator, we took real-world data from three days, from Monday July 10, 2023 12:00 AM PST, to Thursday July 13, 2023 12:00 AM PST. The first two days were used as a train set, and the third day as validation of the calibrated performance on unseen data, as can be seen in Table 2. All times are given in US Pacific, the local time of the real building.

**Calibration Procedure** We ran hyperparameter tuning for 4000 iterations, with the aim of optimizing the TS-MAE, as outlined in equation 1, over the train data. We reviewed the physical constants that yielded the lowest simulation error from calibration. Densities, heat capacities, and conductivities plausibly matched common interior and exterior building materials. However, the external convection coefficient was higher than under the weather conditions, and likely is compensating for the radiative losses and gains, which were not directly simulated. For details about the hyperparameter tuning procedure, including the parameters varied, the ranges given, and the values found that best minimized the calibration metric, see Appendix F.

**Calibration Results** In Table 2, we present the predictive results of our calibrated simulator, on $N$-step prediction, for the train scenario, where $N = 576$, representing a two day predictive window, and the test scenario, where $N = 288$, representing a one day window. We calculated the TS-MAE, as defined in equation 1. We show results for the hyperparameters that best fit the train set, as well as for an uncalibrated simulator as a baseline. At no point was the validation data ever provided to

the tuning process. Note that the validation period is half the duration of the train period, so a lower error does not mean we are performing better than on the train data.

Table 2: Training and test data scenarios

| SPLIT | LENGTH | START | END | CALIBRATED $\epsilon$ | UNCALIBRATED $\epsilon$ |
|-------|--------|-------|-----|-----------|-------------|
| TRAIN | 48 HRS | 2023-07-10 12AM | 2023-07-12 12AM | $0.717\,°C$ | $1.971\,°C$ |
| VAL. | 24 HRS | 2023-07-12 12AM | 2023-07-13 12AM | $0.566\,°C$ | $1.618\,°C$ |

As indicated in Table 2, our tuning procedure drifts only $0.56 \circ C$ on average over a 24-hour period on the validation set.

**Visualizing Temperature Drift Over Time** Figure 4 illustrates temperature drift over time for the training scenario. At each time step, we calculate the spatial temperature for all sensors in both the real building and simulator, and present them as side-by-side boxplot distributions for comparison. Figure 5 shows the same for the validation scenario.

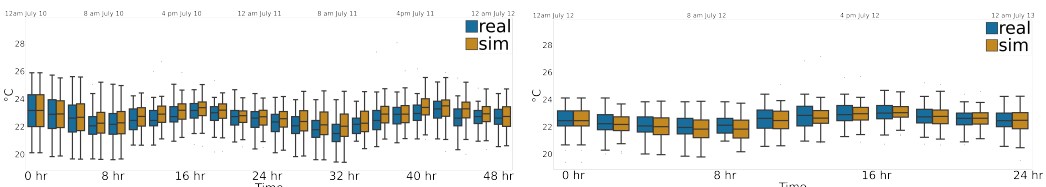

Figure 4: Drift Over 48 hrs on Train Set        Figure 5: Drift Over 24 hrs on Validation Set

Here we can see that our simulator temperature distribution maintains a minimal drift from the real world, although it does seem a bit less reactive to daily fluctuation patterns, which may be the result of the lack of a radiative heat transfer model.

**Visualizing Spatial Errors** Figure 6 illustrates the results of this predictive process over a 24-hour period, on the validation data. It displays a heatmap of the spatial temperature difference throughout the building, between the real world and simulator, after 24 hours of the simulator making predictions. The ring of blue around the building indicates that our simulator is too cold on the perimeter, which implies that the heat exchange with the outside is happening more rapidly than it would in the real world. The inside of the building, at least on the first floor, contains significant amounts of red, indicating that despite the simulator perimeter being cooler than the real world, the inside is warmer. This implies that our thermal exchange within the building is not as rapid as that of the real world. We suspect that this may be because our simulator does not have a radiative heat transfer model. Lastly, there is a large amount of white in this image, indicating that for the most part, even after 24 hours of making predictions on the validation data, our calibration process was successful and the fidelity remains high. For more visuals of spatial errors, see appendix G.

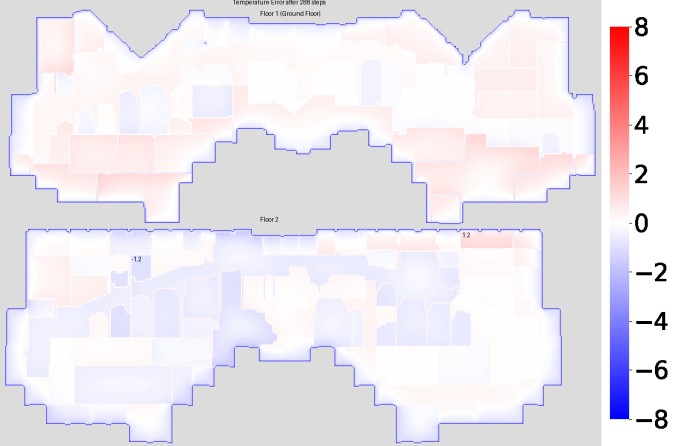

Figure 6: Visualization of simulator drift after 24 hours, on the validation data. The image is a heat map representing the temperature difference between the simulator and the real world, with red indicating the simulator is hotter, blue indicating it is colder, and white indicating no difference. The zone with the max and min temperature difference are indicated by displaying above them the difference.

## 8 DEMONSTRATION BENCHMARKING RESULTS

While we believe our benchmark will be useful for a variety of tasks, such as further use of the data to calibrate the simulator, in this section we highlight results on three important tasks that our suite is well suited to: training an RL agent on the simulator, training a time-series regression model to predict the real world data, and training an RL agent on the real data directly.

**Training a Reinforcement Learning Agent on the Simulator** To demonstrate the usefulness of our calibrated simulator on generating an improved policy, we used Soft Actor Critic (SAC) algorithm (Haarnoja et al., 2018) to train an agent, and then compared our agent with the baseline performance of running the policy currently used in the real building. Both actor and critic were feedforward networks. We ran hyperparameter tuning, again using the method from Golovin et. al. (Golovin et al., 2017), to choose the dimensionality of the critic network and actor network, the batch size, the critic learning rate and actor learning rate, and $\gamma$. We recorded the actor loss, critic loss, alpha loss, and return, over a two day period. The agents trained for 4,000 iterations. Using the $R_{3C}$ reward, the baseline over this two day period had a return of -12.9, and our best agent had an improved return of -11.9, an 8% improvement over the baseline, as show in Table 3. For further training details, and an in depth performance comparison between the learned policy and the baseline, including a breakdown on setpoint deviation, carbon emissions, electrical energy, and natural gas energy, see Appendix H.

Table 3: Policy Comparison

| POLICY | RETURN |
|---|---|
| BASELINE | -12.9 |
| SAC | -11.9 |

**Training a Learned Dynamics Model** Another important task is to use a sequence model to learn to predict the real world data, effectively learning a dynamics model that can then be used in turn in place of the simulator to train an agent. To demonstrate this approach, we trained an encoder-decoder LSTM(Hochreiter, 1997) to model the building dynamics. The model takes in a historical sequence of length $N$ and outputs a prediction sequence of length $M$. At each timestep $t$ in the sequence, the model is given an observation $O_t$, action taken by the policy $A_t$, and auxiliary state features (such as time of day and weather, that are useful as inputs but need not be predicted) $U_t$, and for future timesteps, the model is trained to predict future observations, as well as future reward information (based on predicted energy use and carbon emissions) $E_t$. We evaluated this model by comparing its predictions with the real world data over a three week period, finding that it achieved strong performance and successfully modeled many building dynamics. For detailed architecture diagrams, training information and performance analysis, see Appendix I.

**Training a Reinforcement Learning Agent on Real Data** Building directly off of the above, we also trained an RL agent on the learned dynamics model, demonstrating the ability to learn a policy directly from data without involving the simulator. Like the simulator SAC agent, we were able to learn a policy that improved upon the baseline. For detailed analysis of this policy, see Appendix J.

## 9 LIMITATIONS AND CONCLUSION

The biggest limitation of our benchmark is that all buildings are located in California. We intend to remedy this in the near future by adding more buildings. Another limitation is that we only include data from a one year duration, and in the future we may add longer sequences, for year over year analysis. Our simulator also lacks a radiative heat model, and we hope further work can add this. In addition, our calibration focused on temperature, but in future work we hope to include energy consumption metrics as part of the calibration procedure.

We present a high quality interactive HVAC Control Suite, with real-world historical data from three buildings, as well as calibrated simulators for each building, and a novel, data-based, simulation calibration procedure. We also show promising initial results on key benchmark tasks. We believe this benchmark will facilitate collaboration, reproducibility, and progress on this problem, making an important contribution towards environmental sustainability.

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

## A  REWARD FUNCTION DETAILS

We call our reward function the 3C Reward, because it is made up of a combination of three facors: Comfort, Cost, and Carbon. The purpose of the reward function is to provide the agent a feedback signal after each action about the quality of the current and past actions performed. We combine the different objectives described in Optimization Problem as a normalized, weighted sum of maintaining comfort conditions, electrical cost, and carbon cost:

$$R_{3C} = u \times C_1 + v \times C_2 + w \times C_3$$

where $C_1$ represents normalized comfort conditions, $C_2$ normalized energy cost and $C_3$ normalized carbon emission. Constants $u$, $v$, $w$ represent operator preferences, allowing them to weight the relative importance of cost, comfort and carbon consumption.

Each value $C_1, C_2, C_3$, is bounded by the range $[-1, 0]$, where worst performance is $-1$ and the ideal performance upper-bound is $0$ Thus the reward function in an agregate is formulated as an approximate regret function, bounded in the range [-1,0], and represents an offset from the best-case where comfort conditions are perfectly maintained, without consuming energy and emitting carbon. Each of the sub functions $C_1, C_2, C_3$ will be elaborated next.

### A.1  COMFORT LOSS FUNCTION ($C_1$)

Besides zone air temperature, other factors such as ventilation, drafts, solar exposure, humidity and air quality affect human comfort and productivity in office buildings. However, for now we are focused solely on temperature as the indicator of the comfort level in the office buildings. As additional sensors are deployed and the other factors are measured, they should be considered in the definition of an enhanced comfort loss function.

Studies have shown that a relationship exists between work performance and temperature. For example, in Seppänen, et al. 2006 (Seppanen et al., 2006), work performance was quantified as the mean time required to complete common office tasks (e.g., text processing, bookkeeping calculations, telephone customer service calls, etc.). Performance was shown to increase gradually with temperatures increasing up to 21-22°C and decreasing at temperatures beyond 23-24°C. Therefore, when temperatures deviate outside setpoints, the comfort loss should also be smooth and monotonically increasing.

Thus, the following rules were selected to govern the comfort loss function:

1. Setpoints define the comfort standards, and no penalty should be applied whenever the zone temperature is within heating and cooling setpoints.

2. Comfort is undefined when the zone is unoccupied: if the zone is unoccupied, comfort loss is zero, regardless of zone temperature.

3. Comfort decays smoothly and monotonically as the temperatures drift from setpoints, and occupants are tolerant to small setpoint deviations. Therefore, small setpoint deviations should have a small comfort penalty, and the penalty should smoothly increase as the deviations increase.

4. Large setpoint deviations should approach a maximum, bounded penalty, where a zone becomes completely intolerable for its occupants.

The comfort loss function represents a bounded penalty term for occupied zones that have zone air temperatures outside of setpoint, and covers three adjacent temperature intervals: below cooling setpoint $T_z < \hat{T}_{heating}$, inside setpoints $\hat{T}_{heating} \leq T_z \leq \hat{T}_{cooling}$, and above cooling setpoint $\hat{T}_{cooling} < T_z$

We propose a logistic sigmoid parameterized by $\lambda$ and $\Delta$ to represent the smooth decay (increase loss) of comfort below the heating and above the cooling setpoints. Parameter $\lambda$ is a stiffness coefficient that affects the slope of the decay and parameter $\Delta$ represents the offset in $°C$ from the set point where halfway loss value (0.5) occurs. Additionally we define a step function $\delta(k) = 1$ when the zone has at least one occupant $(k > 0)$, and $\delta(k) = 0$ otherwise.

$$h_z(T_z, k_z, \hat{T}_{heating}, \hat{T}_{cooling}) = \begin{cases} \frac{\delta(k_z)}{1+e^{-\lambda(T_z - \hat{T}_{heating} + \Delta)}} - 1 & T_z < \hat{T}_{heating}] \\ 0 & \hat{T}_{heating} \leq T_z \leq \hat{T}_{cooling} \\ \frac{-\delta(k_z)}{1+e^{-\lambda(T_z - \hat{T}_{cooling} - \Delta)}} & \hat{T}_{cooling} < T_z \end{cases}$$

The chart below shows the comfort loss curve with common setpoints, where the horizontal axis represents zone air temperature and the vertical axis represents the loss. The heating and cooling setpoints were taken from data recordings.

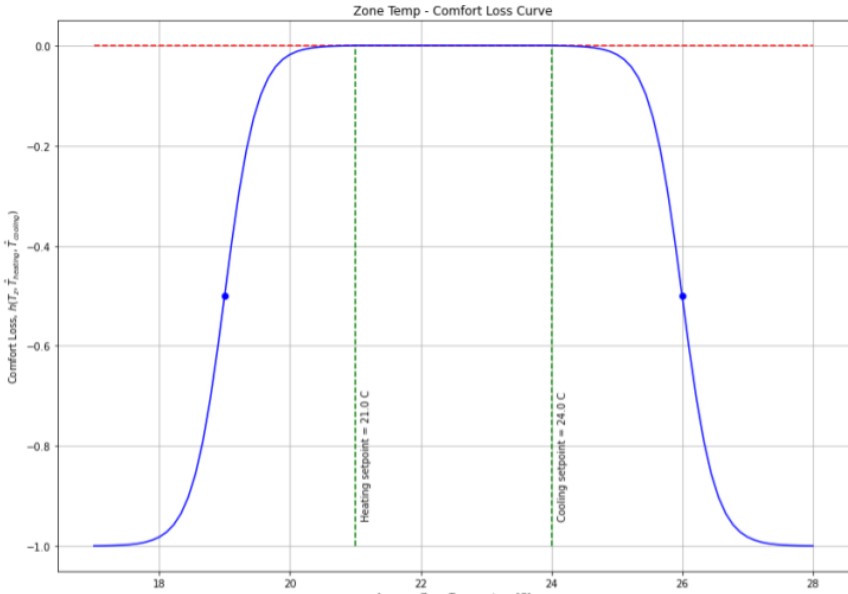

Figure 7: Setpoint Diagram

Finally, we compute the average of all zone comfort losses as the building's overall comfort loss:

$$h_t(S_t) = \frac{1}{|Z|} \times \sum_{z \in Z} h_z(T_z, k, \hat{T}_{heating}, \hat{T}_{cooling})$$

**Live Occupant Feedback** The idea of human feedback shaping the agent's policy may be particularly suitable for the smart buildings project, and has been detailed in Knox and Stone 2009. While not implemented in the initial version of the reward function, the comfort loss function can be extended with an occupant feedback signal reflecting discomfort (e.g., "too hot" or "too cold") in a variety of methods like Mozer 1998 (Mozer, 1998). The agent's goal should be to minimize this type of feedback, and the regret should be increased anytime this feedback signal is received. Suppose one or more occupants in zone $z$, provided a "too cold" feedback signal, $\hat{T}_{heating}$ may be increased by a small amount from the baseline setpoint configuration, and may smoothly return to the baseline smoothly after an appropriate delay.

**Stochastic Occupancy Model** The occupancy signal $k_z$ is the average number of occupants in zone $z$ during a time step $t_i - t_{i-1}$ and is used in computing the comfort loss function described above. Ideally, the occupancy signal is obtained from motion detection sensors or secondary indicators of occupancy, such as wifi signals, badge swipes, calendar appointments, etc. However, a data-driven occupancy signal was not available for the initial dataset, and the following stochastic occupancy model is used instead.

For workdays, we would like model occupancy as a process in the zone where a max number of occupants, $k_{z,max}$ arrive at random times in an arrival window $[\tau_{in,start}, \tau_{in,end}]$, and depart the zone in a departure window $[\tau_{out,start}, \tau_{out,end}]$. The arrivals and departures should occur evenly within the intervals and the expectation of the arrival time should be at the halfway point of the arrival interval:

$\mathbb{E}[\text{occupant arrival time}] = \frac{1}{2}(\tau_{in,end} - \tau_{in,start}) + \tau_{in,start}$

Likewise, the expectation of the departure time should be at the halfway point of the departure interval:

$\mathbb{E}[\text{occupant departure time}] = \frac{1}{2}(\tau_{out,end} - \tau_{out,start}) + \tau_{out,start}$

If the number of timesteps within the arrival and departure intervals is $n_{arrival}$ and $n_{departure}$, this process can be modeled as a geometric distribution where each timestep and occupant is a Bernoulli trial with probabilities:

$P($ occupant arrives — occupant has not yet arrived $) = \frac{2}{n_{arrival}}$ and $P($ occupant departs — occupant has arrived $) = \frac{2}{n_{departure}}$ During holidays and weekends, the zones are not occupied: $k_z = 0$.

## A.2 ENERGY COST FUNCTION ($C_2$)

The energy cost function $C_1(S_t)$ is a normalized, aggregate cost estimate from consuming electrical and natural gas energy during one timestep. The cost function is the ratio of the actual energy used to the maximum energy capacity that ranges between 0: no cost incurred; and 1: maximum cost incurred.

$$C_2(S_t) = -\frac{actual\ energy\ cost}{cost\ at\ max\ energy\ capacity}$$

General energy cost can be calculated as the product of the mean power applied, the time interval, and the cost per unit energy at the time of the interval, where we use $W, \dot{W}$ to represent electrical/mechanical energy, and power, and $Q, \dot{Q}$ to represent thermal energy and power from natural gas. Since all four terms contain the same interval $t_i - t_{i-1}$, they cancel out, allowing us to use power instead of energy. As described above, pumps, blowers, and AC/refrigeration cycles consume electricity and water heaters/boilers consume natural gas. Therefore the total energy and cost is the sum of each energy consumer cost used over the interval:

$$C_2(S_t) = -\frac{(\dot{W}_a + \dot{W}_m + \dot{W}_p) \times p_e(t) + \dot{Q}_g \times p_g(t)}{(\dot{W}_{a,max} + \dot{W}_{m,max} + \dot{W}_{p,max}) \times p_e(t) + \dot{Q}_{g,max} \times p_g(t)}$$

Where $\dot{W}_a$ and $\dot{W}_{a,max}$ are the actual and max electrical power for the AC/refrigeration cycle, $\dot{W}_m$ and $\dot{W}_{m,max}$ are the actual and max electrical power for the blowers/air circulation, $\dot{W}_p$ and $\dot{W}_{p,max}$ are the actual and max pump electrical power, and $\dot{Q}_g$ and $\dot{Q}_{g,max}$ are the actual and max thermal power . Terms $p_e(t)$ and $p_g(t)$ are the electricity and gas price per energy incurred over the interval at time $t$.

The actual power terms in the numerator are estimated from the device observations and the device's fixed parameters using standard HVAC energy conversions. The max power terms in the denominator are derived from device ratings, which define the maximum operating nouns of the device.

## A.3 CARBON EMISSION COST FUNCTION ($C_3$)

Similar to the energy cost function, carbon emission cost function is a function of the electrical and natural gas power used during the interval. The carbon emission cost function $C_3$ is a normalized, aggregate cost estimate from the emission of carbon mass by consuming electrical and natural gas energy during one timestep. The cost function is the ratio of the actual carbon used to the maximum carbon emitted that ranges between 0: no emission cost incurred; and 1: maximum emission cost incurred.

$$C_3(S_t) = -\frac{actual\ carbon\ mass\ emitted}{maximum\ carbon\ emitted}$$

The carbon emission cost is similar to the energy cost function described above, except that we replace the price terms $p_e, p_g$ with emission terms $r_e, r_g$ that convert the power to carbon emission rates.

While the emission rate for natural gas is fairly constant, the emission rate for electricity is dependent on the utility's current renewable energy supply and consumer load during the interval and may fluctuate significantly.

### A.4 IMMEDIATE AND DELAYED REWARD RESPONSES

The reward function is a weighted average of maintaining temperature setpoints in occupied zones, while minimizing energy cost, and minimizing carbon emission. Both energy and carbon emission cost functions provide a low latency response, because actions have an almost immediate effect on the reward. For example, lowering the supply water temperature setpoint will reduce the flow of natural gas to the burner, bringing $\dot{Q}$ down in the next step. However, the effect of increasing water temperature on the comfort loss function may be delayed by multiple time steps, due to the thermal latency in the building. This thermal latency is due to inherent heat capacity and thermal resistance within the building that has a dampening effect on diffusing heat throughout the building. This means that some settings of $u, v, w$ may cause undesirable effects. Experiments with the simulation indicate that too strong weights (e.g., $u + v \geq 0.6$) toward energy cost and/or carbon emission may lead the agent to lower the water temperature, which can cause the VAVs to increase their airflow demand to compensate for a lower supply air temperature, since thermal energy flow is a tradeoff between air mass flow and water heating at the VAV's heat exchanger. Consequently, the increased airflow demand results in a much higher, delayed electrical energy consumption by the blowers to meet the zone airflow demand.

## B PROTO DEFINITIONS

Here, we will elaborate on the exact proto definitions used in the dataset.

Having applied the RL paradigm, the data in our dataset falls under the following categories:

1. **Environment Data** General information about the environment, such as the number of devices and zones, and their names and device types. This is provided once per building environment

2. **Observation Data** The measurements from all devices in the building (VAV's zone air temperature, gas meter's flow rate, etc.), provided at each time step

3. **Action Data** The device setpoint values that the agent wants to set, provided at each timestep

4. **Reward Data** Information used to calculate the reward, as expressed in energy cost in dollars, carbon emission, and comfort level of occupants, provided at each time step

As mentioned above, this data is stored in protos. This section provides the definition of each proto, categorizing them using the four categories above, with examples of each.

### B.1 ENVIRONMENT DATA PROTOS

This is the data that provides, once per environment, details about the environment such as number of devices, and zones, etc. There are two proto definitions:

1. **ZoneInfo:** The `ZoneInfo` message defines thermal spaces or zones in the building and provides zone-to-device association, which enables using the associated VAVs' zone air temperatures to estimate the zone's temperature.

2. **DeviceInfo:** The HVAC devices in the building are defined in the `DeviceInfo` message. Each device exposes a map of `observable_fields` and `action_fields`. The `observable_fields` represent the observable state of the building in native units, and the `action_fields` are available setpoints exposed by the building that the agent may add to its action space. Currently `observable_fields` and `action_fields` are floating point values, but may be expanded to categorical values in the future.

### B.1.1 ZONEINFO DEFINITION

```
1  message ZoneInfo {
2
3
4    enum ZoneType {
5      UNDEFINED = 0;
6      ROOM = 1;
7      FLOOR = 2;
8      OTHER = 10;
9    }
10   // Unique Identifier of the zone.
11   string zone_id = 1;
12   // ID of the building
13   string building_id = 2;
14   // Free-form description of the zone, like microkitchen, office, etc.
15   string zone_description = 3;
16   // Square footage of the zone.
17   float area = 4;  // square meters
18   // Zero to multiple device identifiers associated with this zone, like↩
        VAVs.
19   repeated string devices = 5;
20   // Optional field to describe the type of zone.
21   ZoneType zone_type = 6;
22   // Optional field to indicate the floor of the building.
23   int32 floor = 7;
24 }
```

### B.1.2 ZONEINFO EXAMPLE

```
1  zone_id: "rooms/9028552253"
2  building_id: "buildings/3616672508"
3  zone_description: "US-BLDG-2-C201"
4  devices: "2614466029028994"
5  devices: "2687242320524339"
6  devices: "2640423556868160"
7  zone_type: ROOM
8  floor: 2
```

### B.1.3 DEVICEINFO DEFINITION

```
1  // Details about a specific device in the building.
2  message DeviceInfo {
3    // Device types in smart buildings (official Carson top-level device ↩
        types).
4    enum DeviceType {
5      UNDEFINED = 0;
6      FAN = 1;
7      PMP = 2;
8      FCU = 3;
9      VAV = 4;
10     DH = 5;
11     AHU = 6;
12     BLR = 7;
13     CDWS = 8;
14     CH = 9;
15     CHWS = 10;
16     CT = 11;
17     DC = 12;
18     DFR = 13;
19     DMP = 14;
20     HWS = 15;
21     HX = 16;
```

```
22    MAU = 17;
23    SDC = 18;
24    UH = 19;
25    PWR = 20;
26    GAS = 21;
27    AC = 22;
28    OTHER = 23;
29  }
30
31
32  enum ValueType {
33    VALUE_TYPE_UNDEFINED = 0;
34    VALUE_CONTINUOUS = 1;
35    VALUE_INTEGER = 2;
36    VALUE_CATEGORICAL = 3;
37    VALUE_BINARY = 4;
38  }
39
40
41  // Unique device identifier.
42  string device_id = 1;
43  // If applicable, the zone associated with the device (like VAVs).
44  string namespace = 2;
45  string code = 3;
46  string zone_id = 4;
47
48
49  // The type of device, VAV, AHU, etc.
50  DeviceType device_type = 5;
51  // Map of measurement name exposed by the device to the value type.
52  map<string, ValueType> observable_fields = 6;
53  // Map of setpoint name exposed by the device to their value type.
54  map<string, ValueType> action_fields = 7;
55  }
```

### B.1.4 DEVICEINFO EXAMPLE

```
1  device_id: "202194278473007104"
2  namespace: "PHRED"
3  code: "US-BLDG:AHU:AC-2"
4  device_type: AHU
5  observable_fields {
6    key: "building_air_static_pressure_sensor"
7    value: VALUE_CONTINUOUS
8  }
9  observable_fields {
10   key: "building_air_static_pressure_setpoint"
11   value: VALUE_CONTINUOUS
12  }
13  action_fields {
14   key: "building_air_static_pressure_setpoint"
15   value: VALUE_CONTINUOUS
16  }
17  action_fields {
18   key: "cooling_percentage_command"
19   value: VALUE_CONTINUOUS
20  }
21  action_fields {
22   key: "exhaust_air_damper_percentage_command"
23   value: VALUE_CONTINUOUS
24  }
```

## B.2    OBSERVATION DATA PROTOS

This includes the measurements from all devices in the building (VAV's zone air temperature, gas meter's flow rate, etc.), provided at each time step. There are two proto definitions:

1. **ObservationRequest**
2. **ObservationResponse**

To acquire the latest building state, at each timestep the building accepts an `ObservationRequest` and returns an `ObservationResponse`. The `ObservationRequest` contains a UTC timestamp of the requested observation, and list of `SingleObservationRequests`. Each `SingleObservationRequest` is a tuple of the `device_id` and the `measurement_name` that must match with a device and an `observable_field` in one of the `DeviceInfos` exposed by the building. The building returns an `ObservationResponse` that contains the UTC timestamp from the building, the original `ObservationRequest`, and a list of `SingleObservationResponses`. Each `SingleObservationResponse` contains the associated `SingleObservationRequest`, the validity time of the measurement/observation, a boolean validity indicator, and the observation, in native units, as a continuous, integer, categorical, binary or string value.

### B.2.1    OBSERVATIONREQUEST DEFINITION

```
1  // Agent's request to get the current observation vector.
2  message ObservationRequest {
3   // UTC timestamp when the agent generated the request.
4   google.protobuf.Timestamp timestamp = 1;
5   // One or more individual requests.
6   repeated SingleObservationRequest single_observation_requests = 2;
7  }
8
9
10 // A request to get a single measurement from a specific sensor.
11 message SingleObservationRequest {
12  // Unique device identifier.
13  string device_id = 1;
14  // Name of the sensor, e.g., zone_air_temperature.
15  string measurement_name = 2;
16 }
```

### B.2.2    OBSERVATIONREQUEST EXAMPLE

```
1  timestamp {
2    seconds: 1682649309
3    nanos: 942662000
4  }
5  single_observation_requests {
6    device_id: "202194278473007104"
7    measurement_name: "supply_fan_speed_frequency_sensor"
8  }
9  single_observation_requests {
10   device_id: "202194278473007104"
11   measurement_name: "mixed_air_temperature_sensor"
12 }
13 single_observation_requests {
14   device_id: "202194278473007104"
15   measurement_name: "outside_air_flowrate_setpoint"
16 }
17 single_observation_requests {
18   device_id: "202194278473007104"
19   measurement_name: "supply_air_temperature_sensor"
20 }
```

### B.2.3 OBSERVATIONRESPONSE DEFINITION

```
1  // Building's response to an observation request message.
2  message ObservationResponse {
3    google.protobuf.Timestamp timestamp = 1;
4    ObservationRequest request = 2;
5    repeated SingleObservationResponse single_observation_responses = 3;
6  }
7
8
9
10
11  // Response for a single observation request.
12  message SingleObservationResponse {
13    // The validity time in UTC of the measurement.
14    google.protobuf.Timestamp timestamp = 1;
15    // Original request.
16    SingleObservationRequest single_observation_request = 2;
17    // Validity flag on the observation.
18    bool observation_valid = 3;
19    // Actual observed/measured value.
20    oneof observation_value {
21      float continuous_value = 4;
22      int32 integer_value = 5;
23      string categorical_value = 6;
24      bool binary_value = 7;
25      string string_value = 8;
26    }
27  }
```

### B.2.4 OBSERVATIONRESPONSE EXAMPLE

```
1  timestamp {
2    seconds: 1681110000
3  }
4  request {
5    timestamp {
6      seconds: 1682649309
7      nanos: 942662000
8    }
9    single_observation_requests {
10     device_id: "202194278473007104"
11     measurement_name: "supply_fan_speed_frequency_sensor"
12   }
13   single_observation_requests {
14     device_id: "202194278473007104"
15     measurement_name: "mixed_air_temperature_sensor"
16   }
17   single_observation_requests {
18     device_id: "202194278473007104"
19     measurement_name: "outside_air_flowrate_setpoint"
20   }
21 single_observation_responses {
22   timestamp {
23     seconds: 1681109783
24     nanos: 299000000
25   }
26   single_observation_request {
27     device_id: "202194278473007104"
28     measurement_name: "supply_fan_speed_frequency_sensor"
29   }
30   observation_valid: true
31   continuous_value: 0.0
32 }
```

```
33  single_observation_responses {
34    timestamp {
35      seconds: 1681109783
36      nanos: 299000000
37    }
38    single_observation_request {
39      device_id: "202194278473007104"
40      measurement_name: "mixed_air_temperature_sensor"
41    }
42    observation_valid: true
43    continuous_value: 290.3909912109375
44  }
45  single_observation_responses {
46    timestamp {
47      seconds: 1681109783
48      nanos: 299000000
49    }
50    single_observation_request {
51      device_id: "202194278473007104"
52      measurement_name: "outside_air_flowrate_setpoint"
53    }
54    observation_valid: true
55    continuous_value: 8.825417518615723
56  }
```

### B.3 ACTION DATA PROTOS

This consists of the device setpoint values that the agent wants to set, provided at each timestep. There are two relevant protos:

1. **ActionRequest**
2. **ActionResponse**

The Environment converts the action from the agent into an `ActionRequest` and sends it to the building. The building applies the request and returns an `ActionResponse`. The `ActionRequest` contains the UTC timestamp from the Environment, and a list of `SingleActionRequests`, one for each setpoint in the agent's action space. Each `SingleActionRequest` contains a tuple of the device_id, setpoint_name, and requested setpoint_value, in native units. The device_id must match with one of the device_ids in the `DeviceInfos`, and the setpoint_name must match with one of the action_fields of the associated device. The `ActionResponse` contains the building's UTC timestamp, the original `ActionRequest`, and a list of `SingleActionResponses`, one associated with each `SingleActionRequest`. The `SingleActionResponse` contains the associated `SingleActionRequest`, a response type enumeration, and a string for additional information.

#### B.3.1 ACTIONREQUEST DEFINITION

```
1  // Agent's request to the building with an action.
2  message ActionRequest {
3    // The UTC timestamp that the agent initiated the request.
4    google.protobuf.Timestamp timestamp = 1;
5    // One or more action requests to be performed.
6    repeated SingleActionRequest single_action_requests = 2;
7  }
8
9  // An action request to assign a value to one setpoint on one device.
10 message SingleActionRequest {
11   // The device being commanded.
12   string device_id = 1;
13   // Actual setpoint to be changed, like zone_air_temperature_setpoint.
```

```
14  string setpoint_name = 2;
15  oneof setpoint_value {
16    float continuous_value = 3;
17    int32 integer_value = 4;
18    string categorical_value = 5;
19    bool binary_value = 6;
20    string string_value = 7;
21  }
22 }
```

### B.3.2 ACTIONREQUEST EXAMPLE

```
1 timestamp {
2    seconds: 1682649309
3    nanos: 942662000
4 }
5 single_action_requests {
6    device_id: "12945159110931775488"
7    setpoint_name: "supply_air_static_pressure_setpoint"
8    continuous_value: 186.8100128173828
9 }
10 single_action_requests {
11    device_id: "12945159110931775488"
12    setpoint_name: "supply_air_temperature_setpoint"
13    continuous_value: 294.2592468261719
14 }
15 single_action_requests {
16    device_id: "13761436543392677888"
17    setpoint_name: "supply_water_temperature_setpoint"
18    continuous_value: 310.9259338378906
19 }
20 single_action_requests {
21    device_id: "13761436543392677888"
22    setpoint_name: "differential_pressure_setpoint"
23    continuous_value: 82737.09375
24 }
25 single_action_requests {
26    device_id: "12945159110931775488"
27    setpoint_name: "supervisor_run_command"
28    continuous_value: -1.0
29 }
30 single_action_requests {
31    device_id: "14409954889734029312"
32    setpoint_name: "supervisor_run_command"
33    continuous_value: -1.0
34 }
```

### B.3.3 ACTIONRESPONSE DEFINITION

```
1 // Building's response to an action request.
2 message ActionResponse {
3   // UTC timestamp of the building's response.
4   google.protobuf.Timestamp timestamp = 1;
5   // Original action request.
6   ActionRequest request = 2;
7   // Individual responses for each action.
8   repeated SingleActionResponse single_action_responses = 3;
9 }
10
11
12
13 // Building's response to a single action request.
```

```
14  message SingleActionResponse {
15   enum ActionResponseType {
16     UNDEFINED = 0;
17     // The building accepted the action as requested.
18     ACCEPTED = 1;
19     // The building is processing the request, but has not completed.
20     PENDING = 2;
21     // The action request timed out by request handler.
22     TIMED_OUT = 3;
23     // Request is rejected because the set value is not in an acceptable↩
          range.
24     REJECTED_INVALID_SETTING = 4;
25     // Rejected because the setting is not enabled or available for ↩
          control.
26     REJECTED_NOT_ENABLED_OR_AVAILABLE = 5;
27     // A technician or control function overrode the action.
28     REJECTED_OVERRIDE = 6;
29     // The action was assigned to a device that does not exist.
30     REJECTED_INVALID_DEVICE = 7;
31     // The action was assigned to a valid device that's offline.
32     REJECTED_DEVICE_OFFLINE = 8;
33     UNKNOWN = 9;
34     OTHER = 10;
35   }
36
37
38   SingleActionRequest request = 1;
39   ActionResponseType response_type = 2;
40   // Additional optional information related to the action/response.
41   string additional_info = 3;
42 }
```

### B.3.4 ACTIONRESPONSE EXAMPLE

```
1  timestamp {
2    seconds: 1681110000
3  }
4  request {
5    timestamp {
6      seconds: 1682649309
7      nanos: 942662000
8    }
9    single_action_requests {
10     device_id: "12945159110931775488"
11     setpoint_name: "supply_air_static_pressure_setpoint"
12     continuous_value: 186.8100128173828
13   }
14   single_action_requests {
15     device_id: "12945159110931775488"
16     setpoint_name: "supply_air_temperature_setpoint"
17     continuous_value: 294.2592468261719
18   }
19   single_action_requests {
20     device_id: "13761436543392677888"
21     setpoint_name: "supply_water_temperature_setpoint"
22     continuous_value: 310.9259338378906
23   }}
24 single_action_responses {
25   request {
26     device_id: "12945159110931775488"
27     setpoint_name: "supply_air_static_pressure_setpoint"
28     continuous_value: 186.8100128173828
29   }
30   response_type: ACCEPTED
```

```
31    additional_info: "2023-04-10 06:56:23.299000+00:00 129451591109317754↩
          88"
32  }
33  single_action_responses {
34    request {
35      device_id: "12945159110931775488"
36      setpoint_name: "supply_air_temperature_setpoint"
37      continuous_value: 294.2592468261719
38    }
39    response_type: ACCEPTED
40    additional_info: "2023-04-10 06:56:23.299000+00:00 129451591109317754↩
          88"
41  }
42  single_action_responses {
43    request {
44      device_id: "13761436543392677888"
45      setpoint_name: "supply_water_temperature_setpoint"
46      continuous_value: 310.9259338378906
47    }
48    response_type: ACCEPTED
49    additional_info: "2023-04-10 06:55:33.394000+00:00 137614365433926778↩
          88"
50  }
```

### B.4 REWARD DATA PROTOS

This includes information used to calculate the reward, as expressed in cost in dollars, carbon foot-print, and comfort level of occupants, provided at each time step The Reward protos define the input and output messages for our 3C reward function (Cost Carbon and Comfort), which contains the code that converts them into a single scalar value, a requirement for most RL algorithms. There are two relevant protos:

1. **RewardInfo:** The values that are used as inputs to calculate the reward
2. **RewardResponse:** Containing the scalar reward signal obtained by passing the above functions into our 3C reward function

The building updates the `RewardInfo` at each timestep and provides the reward function necessary inputs to compute the 3C Reward Function. The data contained in the `RewardInfo` is bounded by the step's interval from `start_timestamp` to `end_timestamp` in UTC. The `RewardInfo` has mean energy rate estimates (i.e. power in Watts) that can be treated as constants over the interval. Given the interval and a constant rate value over the interval, the reported power in Watts can be easily converted into energy in kWh. The `RewardInfo` contains maps of three types of specialized data structures:

- The `ZoneRewardInfo` message provides information about the zone air temperature measurements, temperature setpoints, airflow rate and setpoint, and average occupancy for the time step. Each instance is indexed by its unique zone ID.
- The `AirHandlerRewardInfo` message describes the combined electrical power in W use of the intake/exhaust blowers, and the electrical power in W of the refrigeration cycle. Since a building may have more than one air handler, the air handler objects are values in a map keyed by the air handlers' device IDs.
- The `BoilerRewardInfo` contains the average electrical power in W used by the pumps to circulate water through the building, and the average natural gas power in W used to heat the water in the boiler. Since there may be more than one hot water cycle in the building, each `ZoneRewardInfo` is placed into a map keyed by the hot water device's ID.

The reward function converts the current `RewardInfo` into the `RewardResponse` for the same interval as the `RewardInfo`. The agent's reward signal is `agent_reward_value`. Since the reward returned to the agent is a function of multiple factors, it is useful for analysis to show the individual components,m such as carbon mass emitted, and the electrical and gas costs for the step.

### B.4.1 REWARDINFO DEFINITION

```protobuf
message RewardInfo {
 // Information about each zone in the time step for computing reward.
 message ZoneRewardInfo {
    // Heating setpoint of the zone at the timestep in K.
    float heating_setpoint_temperature = 1;

    // Cooling setpoint of the zone at the timestep in K.
    float cooling_setpoint_temperature = 2;

    // Average zone air temperature measured in the zone in K.
    float zone_air_temperature = 3;

    // Setpoint for air flow ventilation in the zone in m^3/s.
    float air_flow_rate_setpoint = 4;

    // Actual ventilation air flow in the zone in m^3/s.
    float air_flow_rate = 5;

    // Average occupancy in the zone over the time step in number of
    // people in the zone.
    float average_occupancy = 6;
 }

 // Information about the air handler energy consumption for computing ↩
     reward.
 message AirHandlerRewardInfo {
    // Cumulative electrical power in W applied to blowers.
    float blower_electrical_energy_rate = 1;

    // Cumulative electrical energy rate applied in W for air ↩
     conditioning. This
    // represents the total power applied for running a refrigeration or
    // heat pump cycles (includes running a compressor and pumps to
    // recirculate refrigerant.).
    float air_conditioning_electrical_energy_rate = 2;
 }

 // Information about the boiler that provides heated water for VAVs.
 message BoilerRewardInfo {
    // Energy rate consumed in W by natural gas for heating water.
    float natural_gas_heating_energy_rate = 1;
    // Cumulative electrical power in W for water recirculation pumps.
    float pump_electrical_energy_rate = 2;
 }

 // Start and end timestamps bound the timestep of the reward ↩
     information.
 google.protobuf.Timestamp start_timestamp = 1;
 google.protobuf.Timestamp end_timestamp = 2;

 // Unique ID of the agent (controller). This should reflect the
 // attributes of the RL models, including the type of algo and its
 // parameters.
 string agent_id = 3;
```

```
62  // Unique ID of the scenario being executed. This should reflect the ←
        details
63  // of the scenario. In simulation, it should identify the canonical ←
        scenario.
64  // In real world, it should define the building and start date/time.
65  string scenario_id = 4;
66
67
68  // Map with zone_id and zone reward info for all zones in the building
69  // under control of the agent. The zone_id could be a unique room ←
        number,
70  // or the specific zone coordinates: (i.e., 'z_i,z_j') from the ←
        simulation.
71  map<string, ZoneRewardInfo> zone_reward_infos = 5;
72
73
74  // Information about the air handlers' energy consumption required to
75  // calculate the reward.
76  map<string, AirHandlerRewardInfo> air_handler_reward_infos = 6;
77
78
79  // Information about the boilers' energy consumption required to ←
        compute the
80  // reward.
81  map<string, BoilerRewardInfo> boiler_reward_infos = 7;
82  }
```

### B.4.2    REWARDINFO EXAMPLE

```
1  start_timestamp {
2    seconds: 1681109700
3  }
4  end_timestamp {
5    seconds: 1681110000
6  }
7  agent_id: "baseline_policy"
8  scenario_id: "baseline_collect"
9  zone_reward_infos {
10   key: "rooms/1000004614278"
11   value {
12     heating_setpoint_temperature: 289.0
13     cooling_setpoint_temperature: 298.0
14     zone_air_temperature: 293.5944519042969
15     air_flow_rate_setpoint: 258.0
16     air_flow_rate: 12.0
17   }
18  }
19  zone_reward_infos {
20   key: "rooms/1000004658174"
21   value {
22     heating_setpoint_temperature: 289.0
23     cooling_setpoint_temperature: 298.0
24     zone_air_temperature: 293.4277648925781
25     air_flow_rate_setpoint: 60.0
26   }
27  }
28  zone_reward_infos {
29   key: "rooms/1000004658175"
30   value {
31     heating_setpoint_temperature: 289.0
32     cooling_setpoint_temperature: 298.0
33     zone_air_temperature: 293.03887939453125
34     air_flow_rate_setpoint: 185.0
```

```
35        air_flow_rate: 4.001242637634277
36      }
37  }
38  zone_reward_infos {
39    key: "rooms/1000004658176"
40    value {
41       heating_setpoint_temperature: 289.0
42       cooling_setpoint_temperature: 298.0
43       zone_air_temperature: 293.53887939453125
44       air_flow_rate_setpoint: 145.0
45       air_flow_rate: 53.0
46    }
47  }
48  air_handler_reward_infos {
49    key: "12945159110931775488"
50    value {
51    }
52  }
53  air_handler_reward_infos {
54    key: "14409954889734029312"
55    value {
56    }
57  }
58  boiler_reward_infos {
59    key: "13761436543392677888"
60    value {
61       pump_electrical_energy_rate: 1527.1470947265625
62    }
63  }
```

### B.4.3 REWARDRESPONSE DEFINITION

```
1  // The return reward signal from the reward function. While the ←
        principal
2  // signal is the agent reward and should be returned to the RL agent, ←
        the
3  // other fields provide useful information for tracking and monitoring.
4  // One EnergyRewardResponse is associated with each EnergyRewardInfo.
5  message RewardResponse {
6
7
8    // Complete reward signal to be returned to the agent.
9    float agent_reward_value = 1;
10
11
12   // Cumulative productivity is measured in USD, and represents the ←
        total
13   // estimated productivity of the building.
14   float productivity_reward = 2;
15
16
17   // Total electrical energy cost estimate in USD.
18   float electricity_energy_cost = 3;
19
20
21   // Total natural gas energy cost in USD.
22   float natural_gas_energy_cost = 4;
23
24
25   // Estimated carbon emitted in kg.
26   float carbon_emitted = 5;
27
28
29   // Estimated carbon cost in USD.
```

```
30   float carbon_cost = 6;
31
32
33   // Productivity weight parameter.
34   float productivity_weight = 7;
35
36
37   // Energy Cost Weight parameter.
38   float energy_cost_weight = 8;
39
40
41   // Carbon emission weight parameter.
42   float carbon_emission_weight = 9;
43
44
45   // Productivity factor (avg labor value of one person-hour).
46   float person_productivity = 10;
47
48
49   // Total average occupancy across all zones.
50   float total_occupancy = 11;
51
52
53   // Reward scale for normalizing the reward
54   float reward_scale = 12;
55
56
57   // Reward shift for normalizing the reward
58   float reward_shift = 13;
59
60
61   // Total productivity regret = max productivity - actual productivity
62   float productivity_regret = 14;
63
64
65   // Normalized productivity regret
66   float normalized_productivity_regret = 15;
67
68
69   // Normalized energy cost =
70   //   combined_energy_cost /
71   //   (max_electricity_energy_cost + max_natural_gas_energy_cost)
72   float normalized_energy_cost = 16;
73
74
75   // Normalized carbon emission =
76   //   combined_carbon_emission /
77   //   (max_electricity_carbon_emission + max_natural_gas_carbon_emission↩
        )
78   float normalized_carbon_emission = 17;
79
80
81   // Start and end timestamps bound the timestep of the reward ↩
        information.
82   google.protobuf.Timestamp start_timestamp = 18;
83   google.protobuf.Timestamp end_timestamp = 19;
84 }
```

### B.4.4 REWARDRESPONSE EXAMPLE

```
1 agent_reward_value: -0.00222194055095315
2 electricity_energy_cost: 0.022907206788659096
3 carbon_emitted: 0.011416268534958363
4 productivity_weight: 0.5
```

```
 5  energy_cost_weight: 0.20000000298023224
 6  carbon_emission_weight: 0.30000001192092896
 7  person_productivity: 300.0
 8  reward_scale: 1.0
 9  normalized_energy_cost: 0.009046462364948
10  normalized_carbon_emission: 0.0013754934770986438
11  start_timestamp {
12    seconds: 1681109700
13  }
14  end_timestamp {
15    seconds: 16811100
```

## C  SIMULATOR DESIGN CONSIDERATION DETAILS

A simulator models the physical system dynamics of the building, devices, and external weather conditions, and can train the control agent interactively, if the following desiderata are achieved:

1. The simulation must produce the same observation dimensionality as the actual real building. In other words, each device-measurement present in the real building must also be present in the simulation.

2. The simulation must accept the same actions (device-setpoints) as the real building.

3. The simulation must return the reward input data described above (zone air temperatures, energy use, and carbon emission).

4. The simulation must propagate, estimate, and compute the thermal dynamics of the actual real building and generate a state update at each timestep.

5. The simulation must model the dynamics of the HVAC system in the building, including thermostat response, setpoints, boiler, air conditioning, water circulation, and air circulation. This includes altering the HVAC model in response to a setpoint change in an action request.

6. The time required to recalculate a timestep must be short enough to train a viable agent in a reasonable amount of time. For example, if a new agent should be trained in under three days (259,200 seconds), requiring 500,000 steps, the average time required to update the building should be 0.5 seconds or less.

7. The simulator must be configurable to a target building with minimal manual effort.

   We believe our simulation system meets all of these listed requirements.

## D  DERIVATION FOR TENSORIZED FINITE DIFFERENCE (FD) EQUATIONS

This appendix describes the method of calculating the flow of heat and the resulting temperatures throughout the building.

### D.1  ASSEMBLING THE ENERGY BALANCE

The fundamental energy balance for a general-purpose closed body is formulated in Equation 3. The first term represents the effects of non-stationary heat dissipation or heat absorption over time over volume of the body. $Q$ represents the energy absorbed or released per unit volume and is a function of the mass and heat capacity of the body. The second term represents thermal flux over the surface of the body, where $\mathbf{n}$ is the unit normal vector of the surface $S$ and $\mathbf{F}$ is the specific energy absorbed or released through the surface. Common modes of thermal flux include conduction, convection, and radiation. The right side of the equation represents the total energy absorbed by the body across the system boundary, or via an external source or sink.

$$\frac{d}{dt} \int_{V(t)} Q dV + \oint_{S(t)} \mathbf{n} \cdot \mathbf{F} dS = \int_{V(t)} P dV \qquad (2)$$

To enable computation, we divide the body into small discrete units, called **Control Volumes** (CV), and iteratively calculate temperature on each on each CV using the method of Finite Differences (FD).

We model three modes of heat transfer into each CV: forced convection, conduction, and external source.

Forced convection $Q^{conv}$ is based on energy exchange by moving air (or any other fluid, in general), and conduction, $Q^{cond}$ is the exchange of energy through solid objects, such as walls. External sources (or sinks) $Q^x$ represent the heating or cooling from external devices, such as electric heating coils, diffusers, etc.

Each CV has the capacity to absorb heat over time, which is expressed as $\frac{dU}{dt}$, governed by its heat capacity, $c$.

These factors allow us to construct an energy balance equation that conserves energy $Q^{in} - Q^{out} = \frac{dU}{dt}$.

We assume that the ceilings and floors are adiabatic, fully insulated, not allowing any heat exchange. This reduces the problem to a 2D problem, with 3D control volumes that can only exchange energy laterally.

Our FD objective is to solve for the temperature at each CV within the building, which presents $N$ unknowns and $N$ equations, where $N$ is the number of CVs in the FD grid.

Rather than creating separate spacial cases in the FD equations for exterior, boundary, and interior CVs, we would like to create a single equation that can be computed across the entire grid. This equation can then be tensorized using the Tensorflow matrix library, and accelerated with GPUs or TPUs.

We label each four interacting surfaces of the CV: left = 1, right = 3, bottom = 2, and top = 4.

Then, for a discrete unit of time $\Delta t$ we specify energy exchange across the surfaces as $Q_1, Q_2, Q_3, Q_4$ and adopt the arbitrary, but consistent convention that energy flows into surfaces 1 and 2, and out of surfaces 3, and 4. (Of course, energy can flow the other direction too, but that will be indicates with a negative value.) Our convention also assumes that external energy flows into the CV.

That allows us to construct the energy balance as:

$$Q^x + Q_1^{cond} + Q_1^{conv} + Q_2^{cond} + Q_2^{conv} - Q_3^{cond} - Q_3^{conv} - Q_4^{cond} - Q_4^{conv} = \frac{dU}{dt} \qquad (3)$$

### D.2 Computing heat transfer via conduction, convection, and thermal absorption

We apply the Fourier's Law of conduction, illustrated in Figure 8, which is the rate of transfer in Watts:

$$\dot{Q}^{cond} = -\frac{kA}{L}\frac{dT}{dt} \qquad (4)$$

Which is approximated over the discrete CV as:

$$\dot{Q}^{cond} \approx -\frac{kA}{L}\frac{\Delta T}{\Delta t} \qquad (5)$$

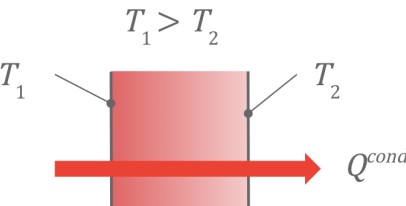

Figure 8: Conduction Heat Transfer

Where $k$ is the thermal conductivity of the material, $A$ is the flux area perpendicular to the flow of heat, $L$ is the distance traveled through the material, $\Delta T$ is the temperature difference in the source and sink, and $\Delta t$ is a discrete time step interval.

We can remove the dot (time derivative) by multiplying by discrete unit time, and converting thermal power (energy per unit time) into energy:

$$Q^{cond} \approx -\frac{kA}{L}\frac{\Delta T}{\Delta t} \times 1 = -\frac{kA}{L}\Delta T \tag{6}$$

Let's orient the conductivity equation along the horizontal ($u$) and the vertical directions ($v$).

For the horizontal heat transfer:

$$Q^{cond}_{1,3} = -\frac{kvz}{u}\Delta T(D.5) \tag{7}$$

And for vertical heat transfer:

$$Q^{cond}_{2,4} = -\frac{kuz}{v}\Delta T \tag{8}$$

Where $z$ is the 3rd dimension size, which is the distance from the floor to the ceiling, and $A = vz$ and $A = uz$ for horizontal and vertical flux surface areas.

This is good for modeling heat exchange through solid objects, but we also need to model the heat exchanges from the outside across the boundary to the interior via forced air convection (i.e., wind).

For convection, we'll apply Newton's Law of Cooling, illustrated in Figure 9 for modeling heat transfer via forced air currents across a surface $A$, perpendicular to the flow of heat as:

$$Q^{cond} = -hA\Delta T \tag{9}$$

The negative sign in Equations 4 - 9 are due to the fact that energy flows in the direction opposite of the temperature gradient, $\Delta T$, i.e., from high to low.

Here, $h$ is the convection coefficient and is a function of the amount of air blowing over the exterior surface of the wall.

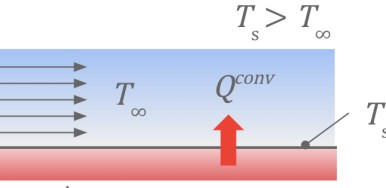

Figure 9: Convection Heat Transfer

We define the three types of CVs:

1. **Exterior CVs** are CVs that represent the ambient weather conditions, such as $T_\infty$ , which are note calculated by the FD calculator, just specified by the current input conditions.

2. **Interior CVs** are CVs where all four sides are adjacent to non-exterior CVs (Figure 10).

3. **Boundary CVs** are CVs that share one or two faces with exterior CVs and one two or three faces with interior CVs. These CVs require special handling, since they represent the transfer of energy between the outside and the inside of the building. Boundary CVs that share two sides with the exterior are **Corner CVs** (Figure 11) and boundary CVs that share only one side with an exterior CV are **Edge CVs** (Figure 12).

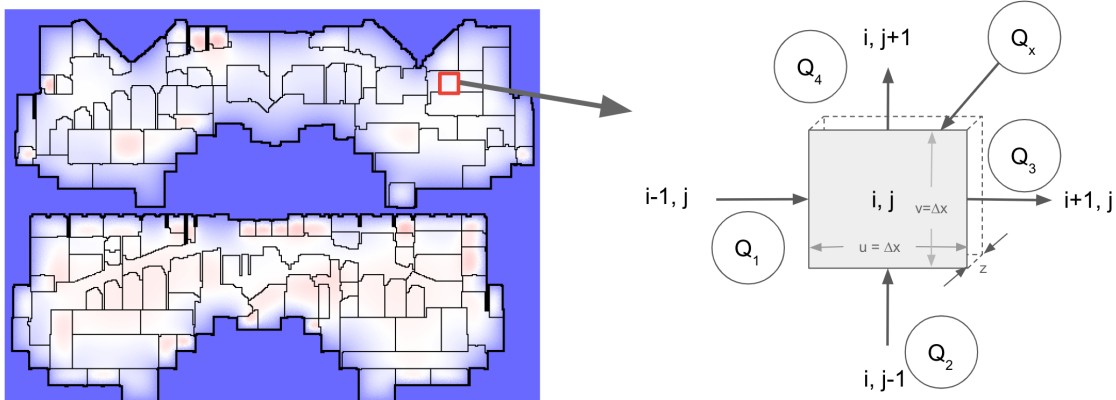

Figure 10: Interior Control Volumes

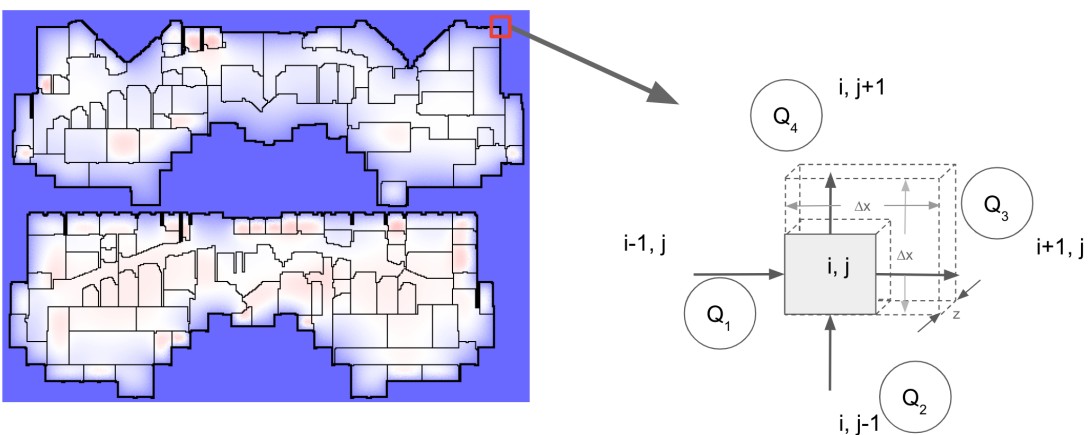

Figure 11: Boundary Corner Control Volumes

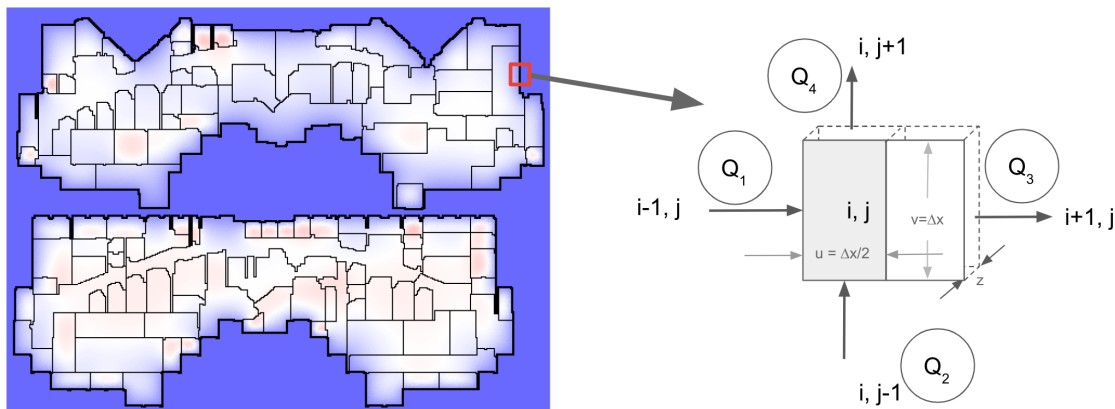

Figure 12: Boundary Edge Control Volumes

The temperatures that are estimated in FD represent the center of the control volume, or its mean. In the case of convection, the temperatures at the exterior surface of the wall us unknown and have to be calculated. Therefore, the center of the Edge CV represents the surface temperature and is split halfway between the outside and inside, where the volume of an edge CV is half of the mass of an interior CV. Similarly, an corner CV is cut in half in both directions, and is one quarter the volume ov an interior CV.

Since we are assuming rectangular CVs, note that $v = v_1 = v_3$, and $u = u_2 = u_4$.

Since outside temperatures and HVAC responses vary, we have a non-stationary thermal system where the flow of energy through the CVs that is not constant. This requires us to evaluate the right-hand term in Equation 3 that allows the volume to absorb or dissipate heat over time, which is governed by the mass $m = \rho V = \rho uvz$, heat capacity $c$ and rate of change of temperature $\frac{dT}{dt}$ .

$$\frac{dU}{dt} = cm\frac{dT}{dt} = c\rho V\frac{dT}{dt} = c\rho uvz\frac{dT}{dt} \tag{10}$$

Equation 10 can be approximated over the small differential CV as:

$$\frac{dU}{dt} \approx c\rho uvz\frac{T_{i,j} - T_{i,j}^{(-)}}{\Delta t} \tag{11}$$

where $T_{i,j}^{(-)}$ is the temperature if the $i,j$ CV at the previous time step and the time step interval is $\Delta t$, which can be treated as a fixed parameter.

### D.3   SOLVING FOR THE TEMPERATURE AT EACH CV

To enable accelerating the calculation using tensor operations, we would like to define a single equation for all CV that do not require (a) conditionals, (b) for loops, or (c) referencing neighboring CVs. That objective will require the construction of a few auxiliary matrices, and every CV will have convection and conduction components that may be disabled with zero-valued convection and conduction coefficients as appropriate.

Combining the Energy Balance in Equation 4 with the conduction and convection equations (Equations 7-10) we can include all terms for all faces on the $i,j$ CV. Our goal is to solve for $T_{i,j}$ which can then be run over multiple sweeps to convergence.

$$Q_x - k_1 vz \frac{T_{i,j} - T_{i-1,j}}{u} - h_1 vz(T_{i,j} - T_\infty) - k_2 uz \frac{T_{i,j} - T_{i,j-1}}{v_2} - h_2 vz(T_{i,j} - T_\infty) +$$

$$+k_3 vz \frac{T_{i+1,j} - T_{i,j}}{u_3} + h_3 vz(T_\infty - T_{i,j}) + k_4 uz \frac{T_{i,j+1} - T_{i,j}}{v_4} + h_4 vz(T_\infty - T_{i,j}) = \quad (12)$$

$$= \frac{c\rho uvz}{\Delta t}\left(T_{i,j} - T_{i,j}^{(-)}\right)$$

Next, we want to solve for temperature $T_{i,j}$ by rearranging the terms, which provides a single equation that can be used to calculate CV temperatures for both boundary and interior CVs.

$$T_{i,j} = \frac{Q_x + vz\left[\frac{k_1}{u}T_{i-1,j} + h_1 T_\infty + \frac{k_3}{u}T_{i+1,j} + h_3 T_\infty\right] + uz\left[\frac{k_2}{v}T_{i,j-1} + h_2 T_\infty + \frac{k_4}{v}T_{i,j+1} + h_4 T_\infty\right] + \frac{c\rho uvz}{\Delta t}T_{i,j}^{(-)}}{vz\left[\frac{k_1}{u} + h_1 + \frac{k_3}{u} + h_3\right] + uz\left[\frac{k_2}{v} + h_2 + \frac{k_4}{v} + h_4\right] + \frac{c\rho uvz}{\Delta t}}$$

$$(13)$$

### D.4 TENSORIZING THE TEMPERATURE ESTIMATE

Equation 13 can be used iterative, but to exploit the acceleration from matrix operations on GPUs and TPUs using the TensorFlow Library, we'll want to reshape the equation slightly for a single tensor pipeline that doesn't iterate over individual CVs.

Furthermore, we can avoid referencing neighboring temperatures $(T_{i-1,j}, T_{i+1,j}, T_{i,j-1}, T_{i,j+1})$ in the pipeline by creating four *shifted* temperature Tensors, $T_1 = \text{shift}(T, 3)$, $T_3 = \text{shift}(T, \text{LEFT})$, $T_2 = \text{shift}(T, \text{UP})$, $T_4 = \text{shift}(T, \text{DOWN})$.

We can also frame oriented conductivity as a Tensors left $K_1$, right $K_3$, below $K_2$, above $K_4$, where:

$$k_{1,i,j} = \begin{cases} k_{i,j} & \text{CVs at } i,j \text{ and } i-1,j \text{ are interior or boundary} \\ 0 & \text{otherwise} \end{cases} \quad (14)$$

$$k_{3,i,j} = \begin{cases} k_{i,j} & \text{CVs at } i,j \text{ and } i+1,j \text{ are interior or boundary} \\ 0 & \text{otherwise} \end{cases} \quad (15)$$

$$k_{2,i,j} = \begin{cases} k_{i,j} & \text{CVs at } i,j \text{ and } i,j-1 \text{ are interior or boundary} \\ 0 & \text{otherwise} \end{cases} \quad (16)$$

$$k_{4,i,j} = \begin{cases} k_{i,j} & \text{CVs at } i,j \text{ and } i,j+1 \text{ are interior or boundary} \\ 0 & \text{otherwise} \end{cases} \quad (17)$$

Note that the conductivity matrix $K$ is a fixed input parameter for the building.

Applying the same reasoning, we can generate four oriented convection Tensors, $H_1, H_2, H_3, H_4$ as:

$$h_{1,i,j} = \begin{cases} h & \text{CV at } i,j \text{ is boundary and CV at } i-1,j \text{ is exterior} \\ 0 & \text{otherwise} \end{cases} \quad (18)$$

$$h_{3,i,j} = \begin{cases} h & \text{CV at } i,j \text{ is boundary and CV at } i+1,j \text{ is exterior} \\ 0 & \text{otherwise} \end{cases} \quad (19)$$

$$h_{2,i,j} = \begin{cases} h & \text{CV at } i,j \text{ is boundary and CV at } i,j+1 \text{ is exterior} \\ 0 & \text{otherwise} \end{cases} \quad (20)$$

$$h_{4,i,j} = \begin{cases} h & \text{CV at } i,j \text{ is boundary and CV at } i,j-1 \text{ is exterior} \\ 0 & \text{otherwise} \end{cases} \tag{21}$$

Note that $h$ is a time-dependent constant that represents the amount of airflow over the surface of the building, assumed to be uniformly applied on all exterior walls of the building.

Finally, we classify each boundary CV as TOP-LEFT CORNER, TOP-RIGHT CORNER, BOTTOM-LEFT CORNER, BOTTOM-RIGHT CORNER or LEFT EDGE, RIGHT EDGE, TOP EDGE, or BOTTOM EDGE in order to form Tensors $U$ and $V$, which are the CV widths and heights.

$$u_{i,j} = \begin{cases} \frac{\Delta x}{2} & \text{CV at } i,j \text{ is BOUNDARY and ANY CORNER or TOP or BOTTOM EDGE} \\ \Delta x & \text{otherwise} \end{cases} \tag{22}$$

$$v_{i,j} = \begin{cases} \frac{\Delta x}{2} & \text{CV at } i,j \text{ is BOUNDARY and ANY CORNER or LEFT or RIGHT EDGE} \\ \Delta x & \text{otherwise} \end{cases} \tag{23}$$

where $\Delta x$ is the fixed horizontal and vertical dimension of an INTERIOR CV.

Now we can complete the Tensor expression of the FD equation:

$$
\begin{aligned}
T = \Big[ Q_x + Vz \left[ K_1 U^{-1} T_1 + H_1 T_\infty + K_3 U^{-1} T_3 + H_3 T_\infty \right] \\
+ Uz \left[ K_2 V^{-1} T_2 + H_2 T_\infty + K_4 V^{-1} T_4 + H_4 T_\infty \right] \\
+ \frac{CPUVz}{\Delta t} T^{(-)} \Big] \\
\cdot \left[ Vz \left[ K_1 U^{-1} + H_1 + K_3 U^{-1} + H_3 \right] + Uz \left[ K_2 V^{-1} + H_2 + K_4 V^{-1} + H_4 \right] + \frac{CPUVz}{\Delta t} \right]^{-1}
\end{aligned}
\tag{24}
$$

For each timestep, we execute Equation 24 as single-step tensor operations until convergence, where the maximum change across all CVs between current and last iteration is less then a conservative lower threshold, $\epsilon \le 0.01°C$

## E   SIMULATOR CONFIGURATION PROCEDURE DETAILS

To configure the simulator, we require two type of information on the building:

1. Floorplan blueprints. This includes the size and shapes of rooms and walls for each floor.
2. HVAC metadata. This includes each device, its name, location, setpoints, fixed parameters and purpose.

We preprocess the detailed floorplan blueprints of the building, and extract a grid that gives us an approximate placement of walls and how rooms are divided. This is done via the following procedure:

1. Using threshold $t$, binarize the floorplan image into a grid of 0s and 1s.
2. Find and replace any large features that need to be removed (such as doors, a compass, etc)
3. Iteratively apply standard binary morphology operations (erosion and dilation) to the image to remove noise from background, while preserving the walls.

4. Resize the image, such that each pixel represents exactly one control volume

5. Run a connected components search to determine which control volumes are exterior to the building, and mark them accordingly

6. Run a DFS over the grid, and reduce every wall we encounter to be only a single control volume thick in the case of interior wall, and double for exterior wall

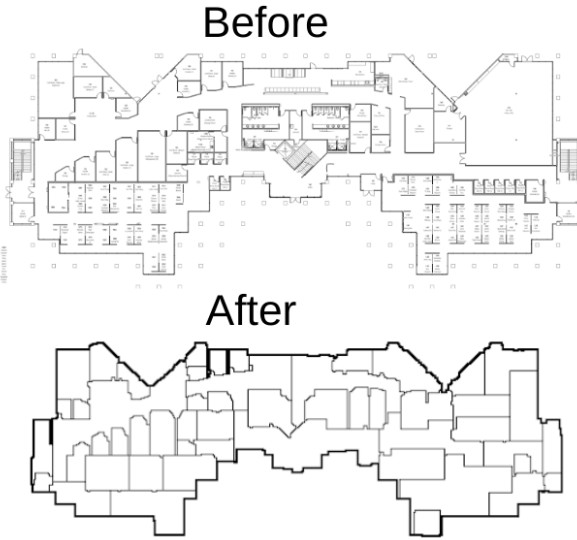

Figure 13: Before and after images of the floorplan preprocessing algorithm

We also employ a simple user interface to label the location of each HVAC device on the floorplan grid. This information is passed into our simulator, and a custom simulator for the new building, with roughly accurate HVAC and floor layout information, is created. This allows us to then calibrate this simulator using the real world data, which will now match the simulator in terms of device names and locations.

We tested this pipeline on SB1, which consisted of two floors with combined surface area of 93,858 square feet, and has 127 HVAC devices. Given floorplans and HVAC layout information, a single technician was able to generate a fully specified simulation in under three hours. This customized simulator matched the real building in every device, room, and structure.

## F    CALIBRATION HYPERPARAMETER TUNING DETAILS

The hyperparameter tuning was performed over a seven day period on 200 CPUs.

Table 4: Thermal properties that were set by the calibration process, with min/max bounds and selected values.

| HYPERPARAMETER | MIN | MAX | BEST |
|---|---|---|---|
| CONVECTION_COEFFICIENT $(W/m^2/K)$ | 5 | 800 | 357 |
| EXTERIOR_CV_CONDUCTIVITY $(W/m/K)$ | 0.01 | 1 | 0.83 |
| EXTERIOR_CV_DENSITY $(kg/m^3)$ | 0 | 3000 | 2359 |
| EXTERIOR_CV_HEAT_CAPACITY $(J/Kg/K)$ | 100 | 2500 | 2499 |
| INTERIOR_WALL_CV_CONDUCTIVITY $(W/m/K)$ | 5 | 800 | 5 |
| INTERIOR_WALL_CV_DENSITY $(kg/m^3)$ | 0.5 | 1500 | 1500 |
| INTERIOR_WALL_CV_HEAT_CAPACITY $(J/Kg/K)$ | 500 | 1500 | 1499 |
| SWAP_PROB | 0 | 1 | 0.003 |
| SWAP_RADIUS | 0 | 50 | 50 |

## G    ADDITIONAL SPATIAL ERROR VISUALIZATIONS

Here we present some other visuals that may be enlightening.

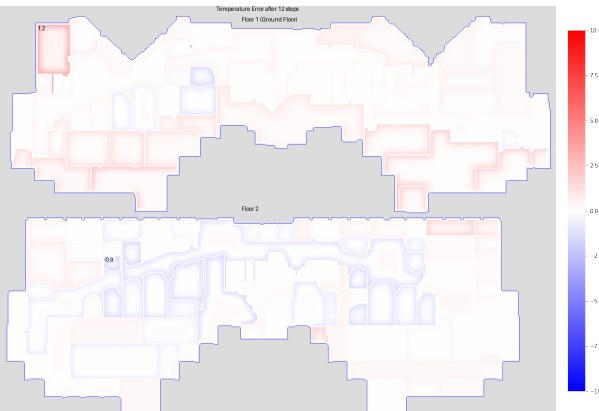

Figure 14: Visualization of simulator drift after only a single hour, on the validation data. As can be clearly seen, at this point there is almost no error.

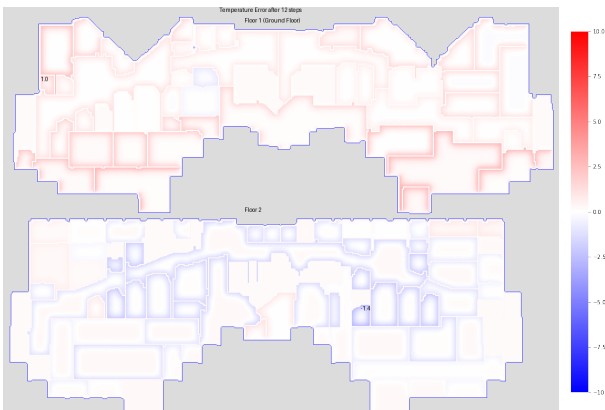

Figure 15: Visualization of simulator drift after only a single hour, on the train data. Again, there is almost no error.

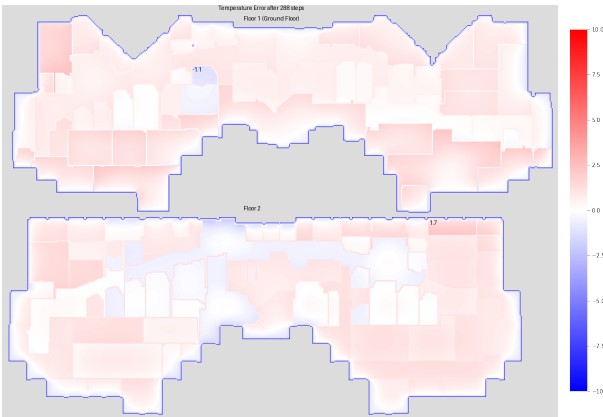

Figure 16: Visualization of simulator drift after one day, on the train data.

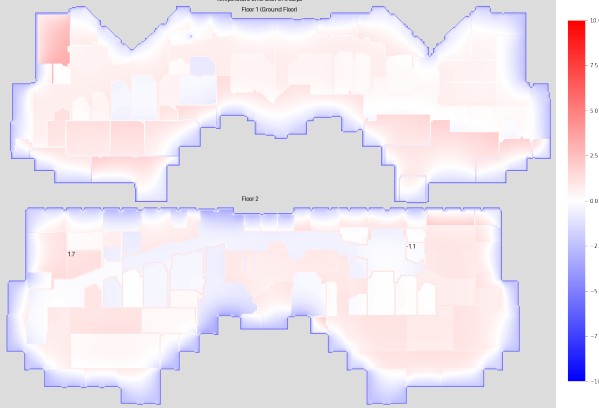

Figure 17: Visualization of simulator drift after two days, on the train data. Interestingly, this looks better than it did after only one day.

## H SIMULATOR SAC AGENT TRAINING DETAILS AND PERFORMANCE ANALYSIS

We will now go into more details on the simulator SAC agent training and performance as compared to the baseline.

Each agent was trained on a single CPU, with the entire training session lasting 6 days. We restricted the action space to supply air and water temperature setpoints. For the observation space, we found that providing the agent with the dozens of temperature sensors was too much noisy information and not useful. Instead, we provided the agent with a histogram, grouping temperatures into $1°$ Celsius bins, ranging from $12°$ to $30°$, and calculating the frequency of each bin. The tallies are then normalized and provided as part of the observation. This led to much better performance.

Figure 18 shows the returns during training.

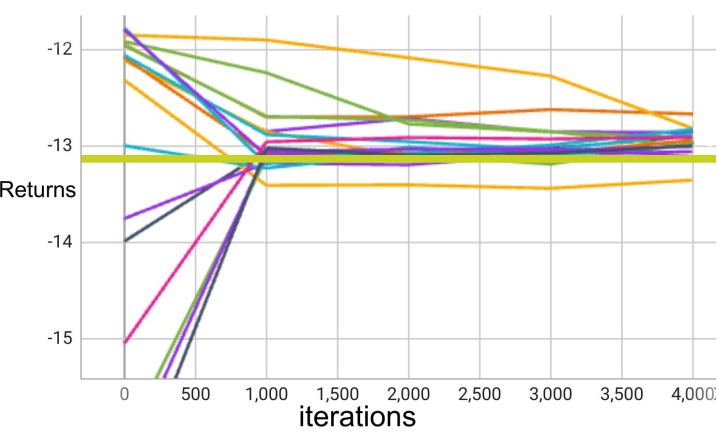

Figure 18: SAC agent Returns of each agent we trained, as well as the baseline in gold, which represents the returns obtained by running the baseline policy currently employed in the real world. As can be clearly seen, most of the agents are able to improve above this policy.

Figure 19 illustrates that the critic, actor, and alpha losses of the various SAC agents converge.

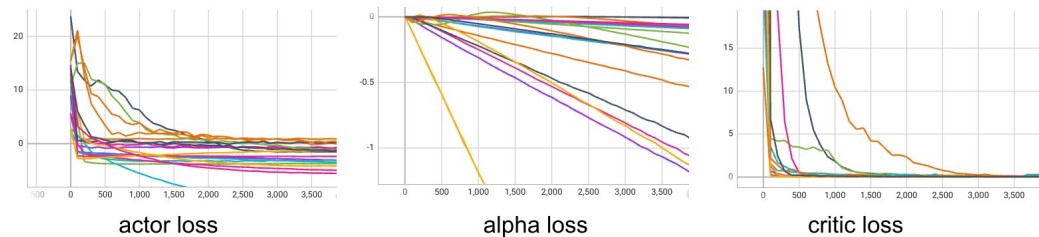

Figure 19: SAC Agent Losses

Our reward function is a weighted, linear combination of the normalized carbon footprint, cost, and comfort levels within the building. While an 8% improvement over the baseline on this scalar reward is significant, we can see the improvements of the SAC agent over the baseline even more clearly when we break down these factors further into physical measures.

For this analysis, we break down the reward into four components that contribute to it, and see how the learned policy compares with the baseline. The components are: setpoint deviation, carbon emissions, electrical energy, and natural gas energy.

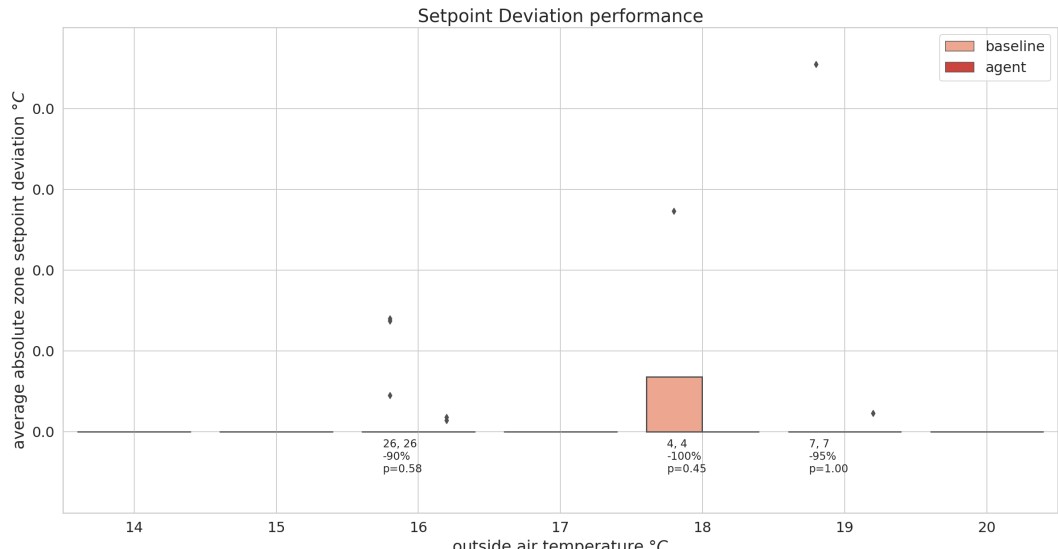

Figure 20: Setpoint Deviation Performance as a function of outside air temperature, which evaluates how well the agent meets comfort conditions compared to the baseline. It is measured as the average number of $°C$ above or below setpoint for all zones in the building. For each outside air degree increment, we include the number of observations for baseline and agent, the percentage change as (baseline - agent) / baseline, and its associated p-score.

Above we display how the baseline and agent compare when it comes to setpoint deviation, the comfort component of the reward function. We show the distribution of deviations grouped by outside air temperatures. While both policies have very minimal setpoint deviation to begin with, the agent strictly improves over the baseline here.

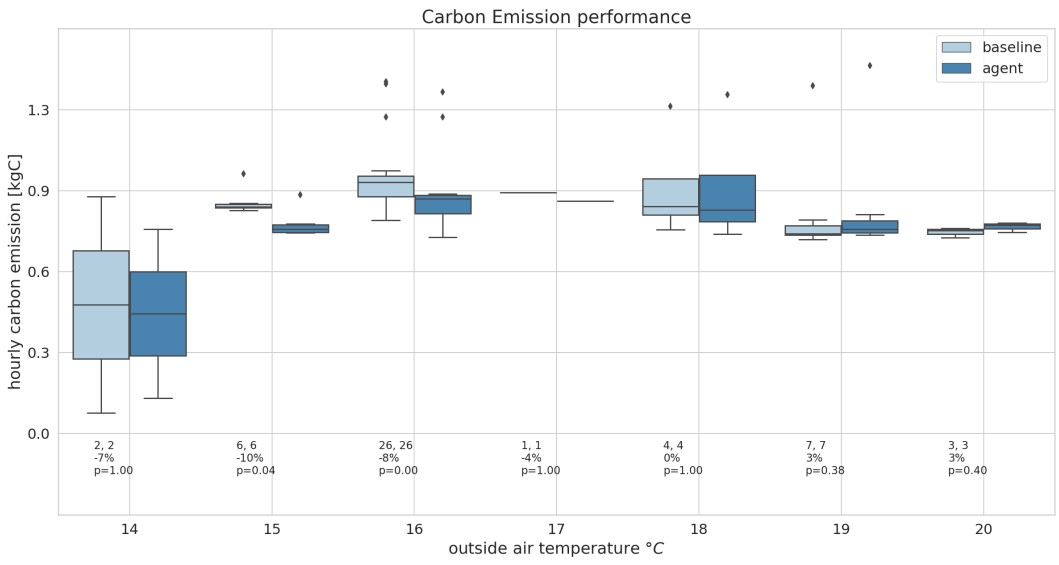

Figure 21: Carbon Emission measures how the agent performs compared to the baseline in terms of the amount of greenhouse gas released from consuming natural gas and electricity. C is combined mass (kgC, or kg Carbon) emitted by non-renewable electricity and natural gas. For each outside air degree increment, we include the number of observations for baseline and agent, the percentage change as (baseline - agent) / baseline, and its associated p-score.

The carbon performance of the agent, as compared with the baseline, is impressive as well. In the temperature range $14°C$ to $18 °C$, the agent is strictly better, and while it is slightly worse for the warmer temperatures, clearly it is a net improvement over the baseline.

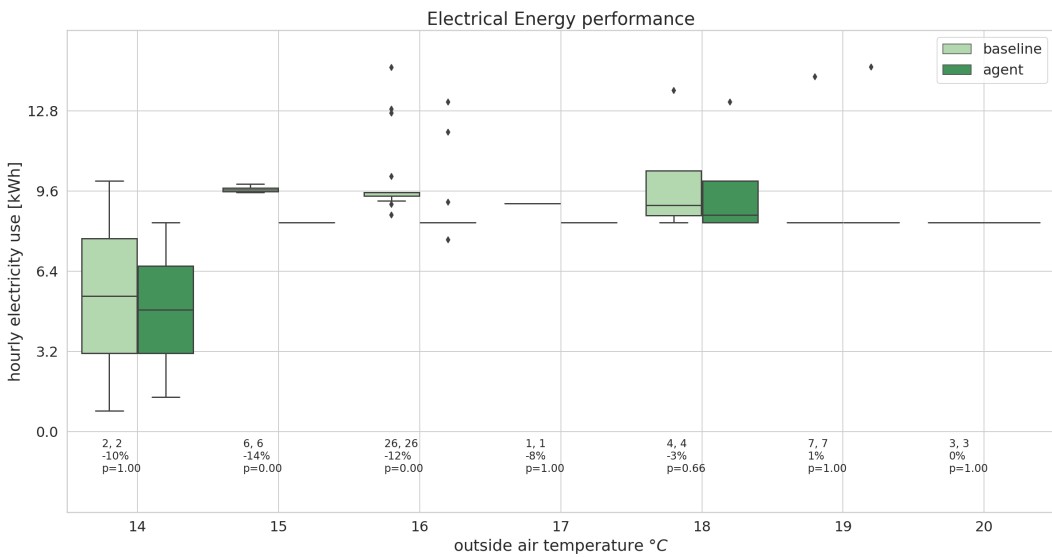

Figure 22: Electrical Energy Performance measured in energy units (kWh) over a fixed interval for both the agent and the baseline policies. For each outside air degree increment, we include the number of observations for baseline and agent, the percentage change as (baseline - agent) / baseline, and its associated p-score.

Once again, when it comes to electric performance, the SAC agent is almost strictly better under all temperature ranges.

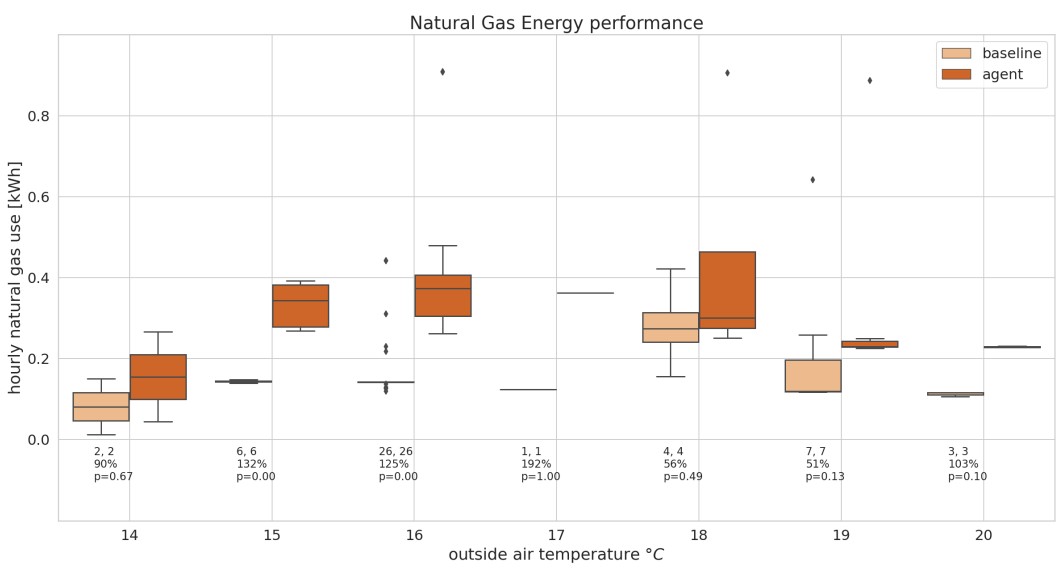

Figure 23: Natural Gas Performance measured in energy units (therm) over a fixed interval for both the agent and the baseline policies. For each outside air degree increment, we include the number of observations for baseline and agent, the percentage change as (baseline - agent) / baseline, and its associated p-score.

Interestingly, the agent converged on a policy that reduced overall carbon emission while increasing natural gas consumption. This is due to the fact that electricity is generated from non-renewable sources and per unit energy, is significantly more expensive than gas.

# I   TRAINING AND EVALUATING A LEARNED DYNAMICS MODEL

Aside from being useful for offline training and for calibrating our simulator, the real world data can also be used to directly learn a regression model that approximates the building dynamics. This model can then be used to train a control agent.

As described in the main paper, to demonstrate this approach, building off of earlier work(Velswamy et al., 2017; Sendra-Arranz & Gutiérrez, 2020; Zou et al., 2020; Zhuang et al., 2023), we trained an LSTM to model the building dynamics. We used an encoder-decoder network, where the model takes in a historical sequence of length $N$ and outputs a prediction sequence of length $M$. At each timestep $t$ in the sequence, the model is given an observation $O_t$, action taken by the policy $A_t$, and auxiliary state features (such as time of day and weather, that are useful as inputs but need not be predicted) $U_t$, and for future timesteps, the model is trained to predict future observations, as well as future reward information (based on predicted energy use and carbon emissions) $E_t$. The LSTM model is shown in Figure 24.

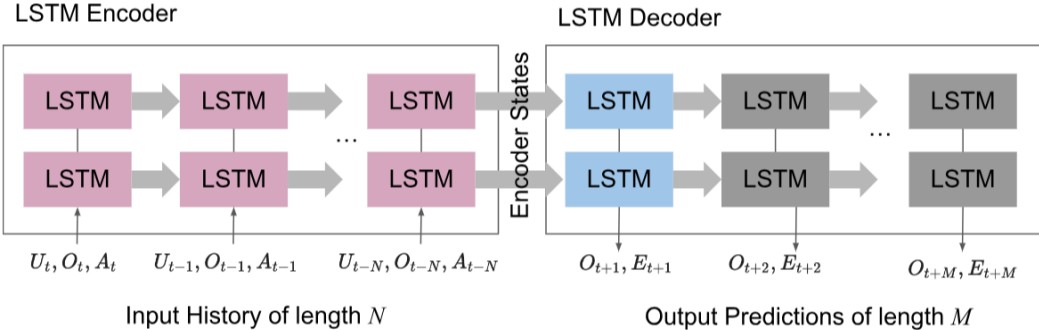

Figure 24: Architecture of LSTM building dynamics model

We then trained the model to predict the next observation for 65 epochs, plotting training and validation loss, as shown in figure 25

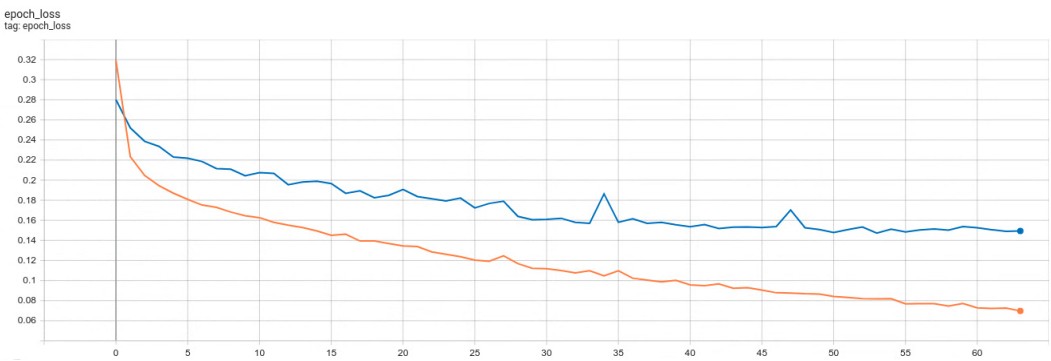

Figure 25: Loss of LSTM building dynamics model, with train loss in orange and validation loss in blue.

However, loss curves alone do not tell the full story of how well our regression model is reconstructing the signal of the dynamics, so we also included additional evaluations. We had the model predict 3 weeks into the future, and then compared the predictions with the ground truth data to ensure the

cyclic patterns of the medians are reproduced. The chart in figure 26 shows 20 measurement time series from the regression models shown in yellow compared to the actual values shown in gray. By inspection, we conclude that the regression building provides good correspondence with the actual real data signals.

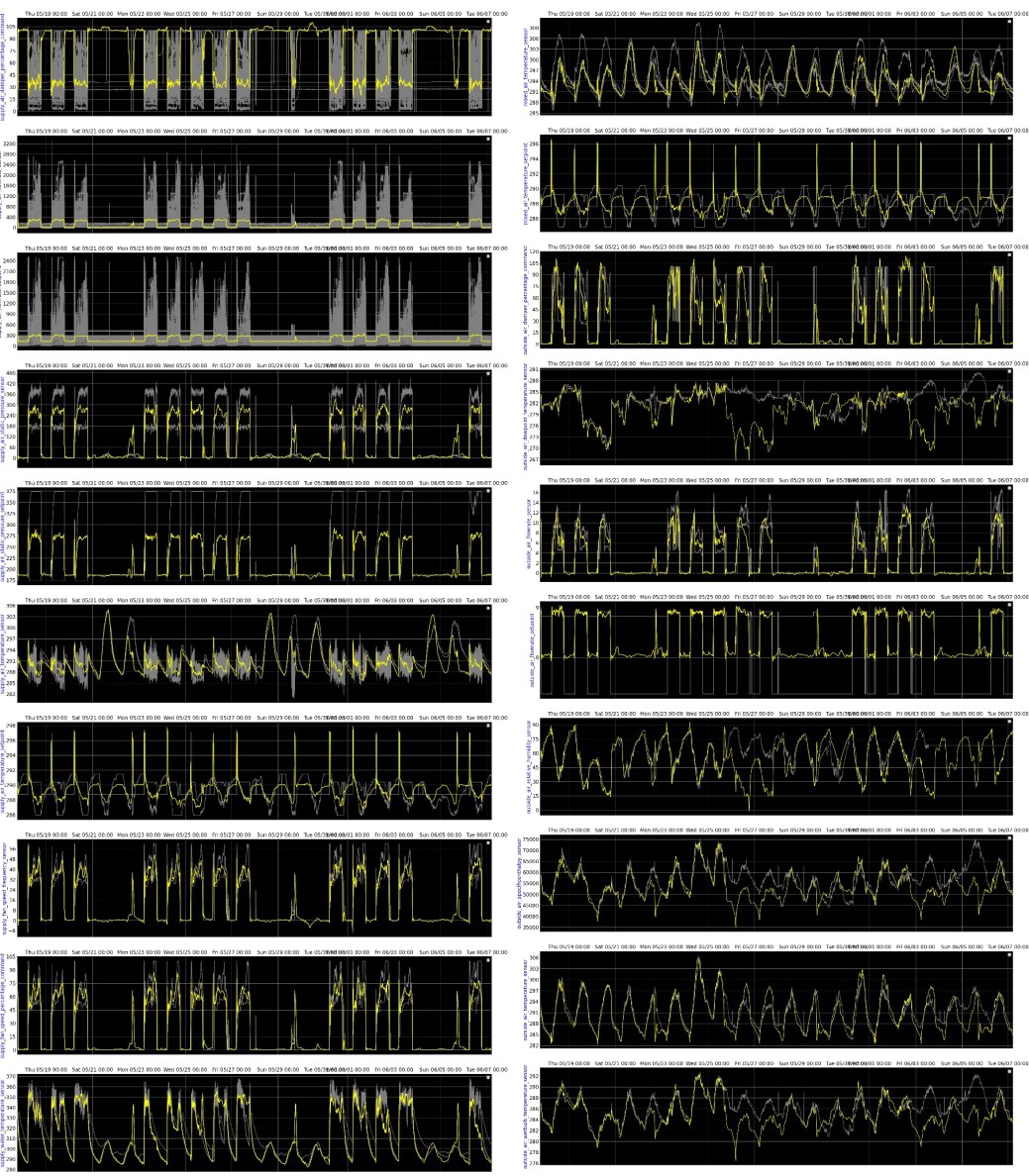

Figure 26: Detailed analysis of learned dynamics as compared to real data.

## J    REAL DATA SAC AGENT TRAINING DETAILS AND PERFORMANCE ANALYSIS

We then trained a SAC agent on the regression environment, much like how we did on the simulator. This gives us a baseline for how to generate a policy purely based on data, without use of the simulator. We used hyper-parameter tuning, and trained 200 agents. The chart in figure 27 shows agent reward progress as the number of trials increased.

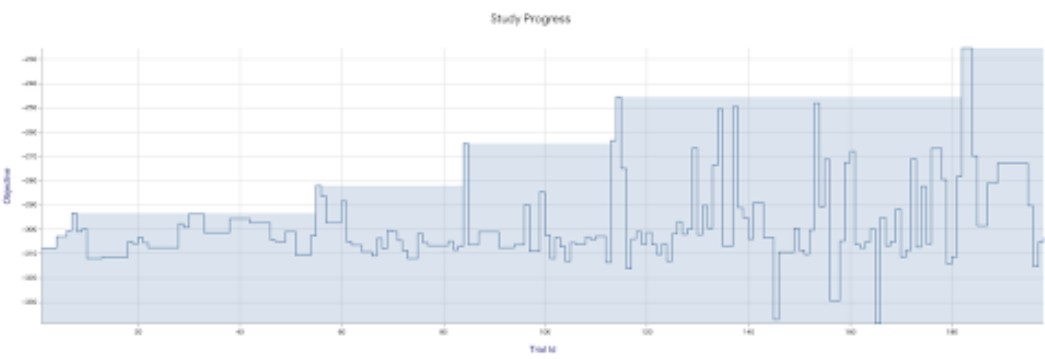

Figure 27: Detailed analysis of learned dynamics as compared to real data.

To compare the learned policy with the baseline, we plotted the two policies in 28. The baseline and agent episode temperature timelines shown below provide a temporal perspective of the environment median zone air temperatures (yellow) and setpoints (white), outside air temperature (blue), and the agent actions on the environment (water temperature setpoints (lime), and air handler temperature setpoints (magenta). While the regression model under baseline policy correctly represents the weekend setpoint ranges, the regression building applies nearly the weekday setpoint ranges when running under agent control. This is likely due to the agent applying setpoints that regression associates with weekday actions, and incorrectly returns a setpoint that is closer to the weekday. For this reason, we do not evaluate the model's performance on weekends. Similar to baseline control, the agent ramps up water temperature (lime) at the beginning of the day. However, the agent tends to maintain the water temperature around 80C for substantially longer than baseline control. At first glance, this may seem counterproductive. However, heat exchange is also based on water flow and air flow. Lower supply water temperatures require more airflow to transfer the same amount of heat. Therefore, higher water temperatures do not necessarily result in higher energy consumption. Also, note that the agent does not drop the water temperature as low as the baseline policy, and the agent tends to apply smoother actions compared to the baseline's rapid oscillation between 40 and 60C. We speculate that one strength of the proposed solution is the agent's ability to discover better and non-intuitive policies that are unlikely to be chosen by human HVAC technicians. The agent also has a different control policy for the air handlers' supply air temperatures, shown in magenta. On one air handler's supply air temperature, the agent tends to operate SB1:AHU:AC 1 at a higher temperature than SB1:AHU:AC 2.

Finally, much like how we did with the simulated agent, we break down the reward into its four components and see how the agent did relative to the baseline on the regression building model.

## J.1 SETPOINT MANAGEMENT PERFORMANCE

The difference in setpoint deviation between agent and baseline was insignificant. However, at 23C the average setpoint deviation was slightly higher, but was still within a narrow window (less than 1/10 C). The setpoint deviation test using the regression model may be slightly optimistic compared to the real building, because the regression model only approximates the zone temperatures with a single median, hiding the larger spread of temperatures throughout the building.

## J.2 CARBON EMISSION PERFORMANCE

In 12 of 19 temperature bins the agent generated less carbon than the baseline. While only two temperature bins (17C, 25C) resulted in confidence greater than 90%, the results indicate a reduction in carbon emission on most of the bins. The agent tends to emit more carbon in the moderate temperature ranges (21, 22C), likely due to a higher setpoint during the day than the baseline. Overall, the agent performs favorably, even though most bins have a low statistical confidence.

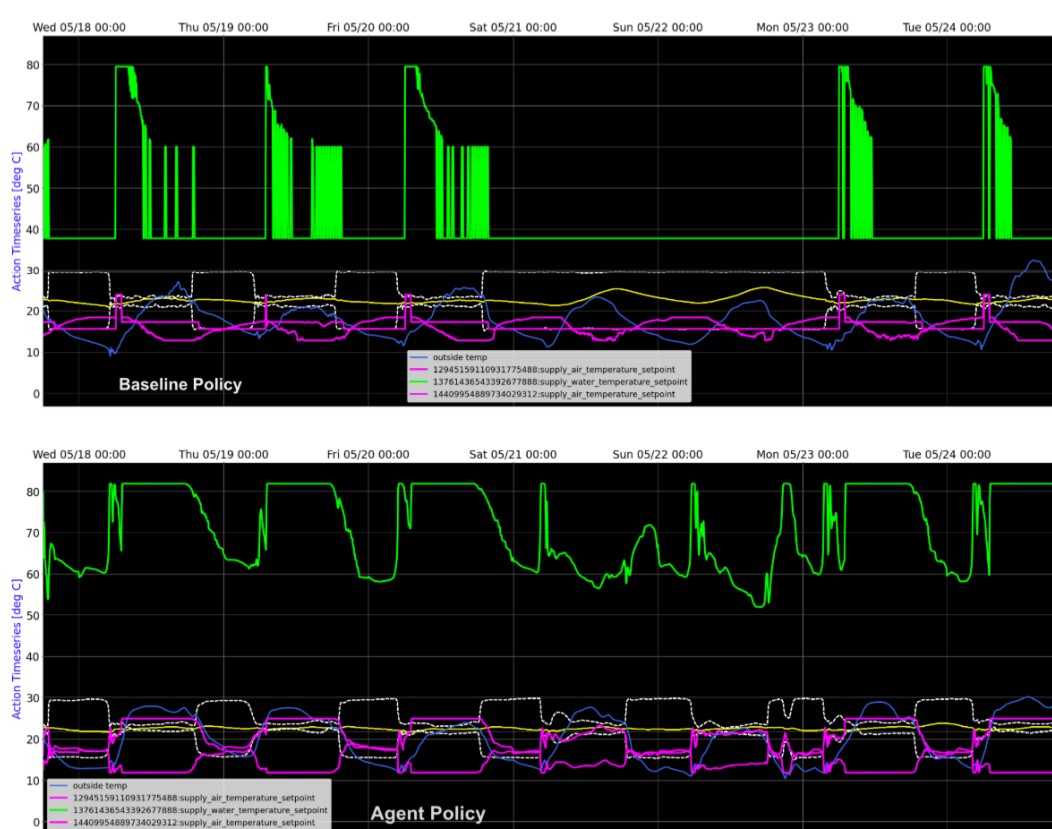

Figure 28: Detailed analysis of learned dynamics as compared to real data.

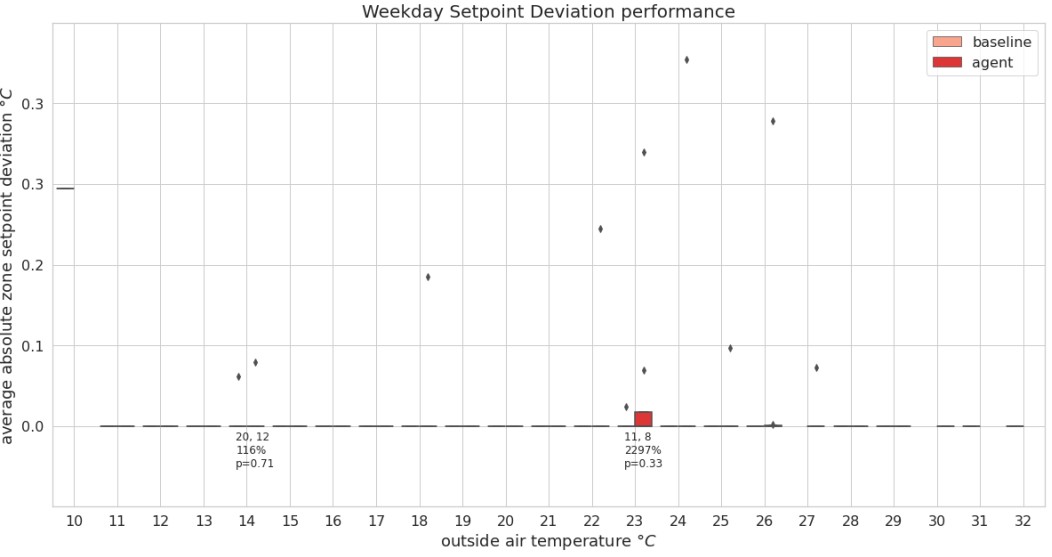

Figure 29: Setpoint Management Performance.

## J.3 ELECTRICAL ENERGY PERFORMANCE

While no temperature bins yielded confidence scores greater than 90%, the agent tends to consume less electricity than the baseline, except for the 21, 22C temperature bins. Under both policies, electricity consumption dramatically increases with outside air temperature.

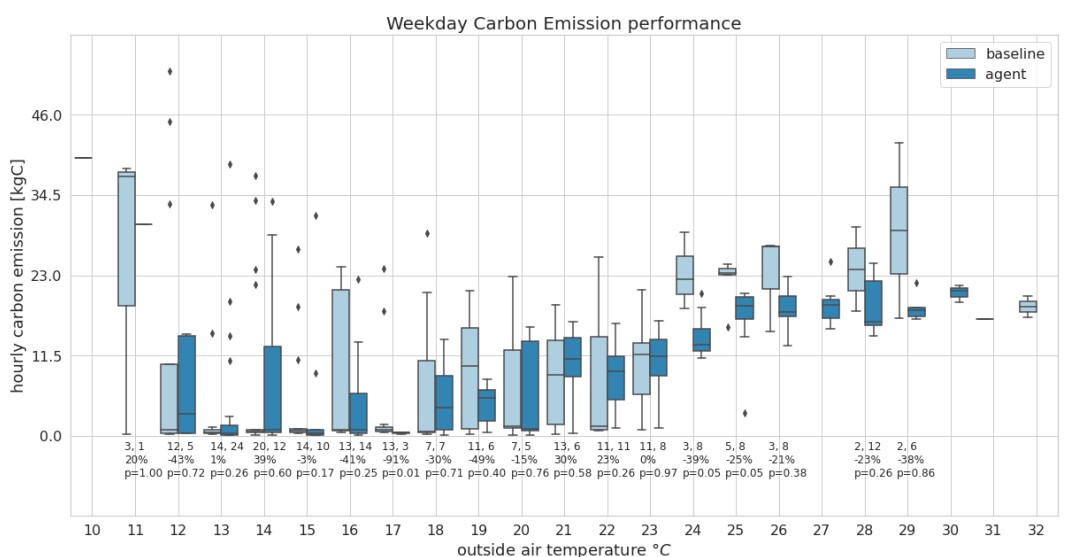

Figure 30: Carbon Emission Performance.

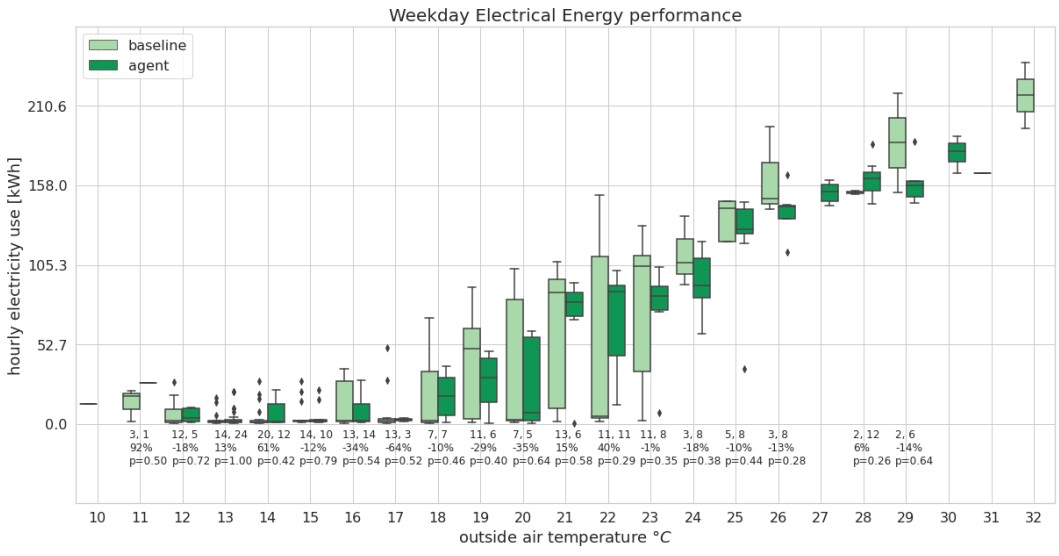

Figure 31: Electrical Energy Performance.

## J.4 NATURAL GAS ENERGY PERFORMANCE

The agent policy tends to consume less natural gas than the baseline policy, even though only three yielded significant reduction with confidence of at least 90% (17, 24, 25C).

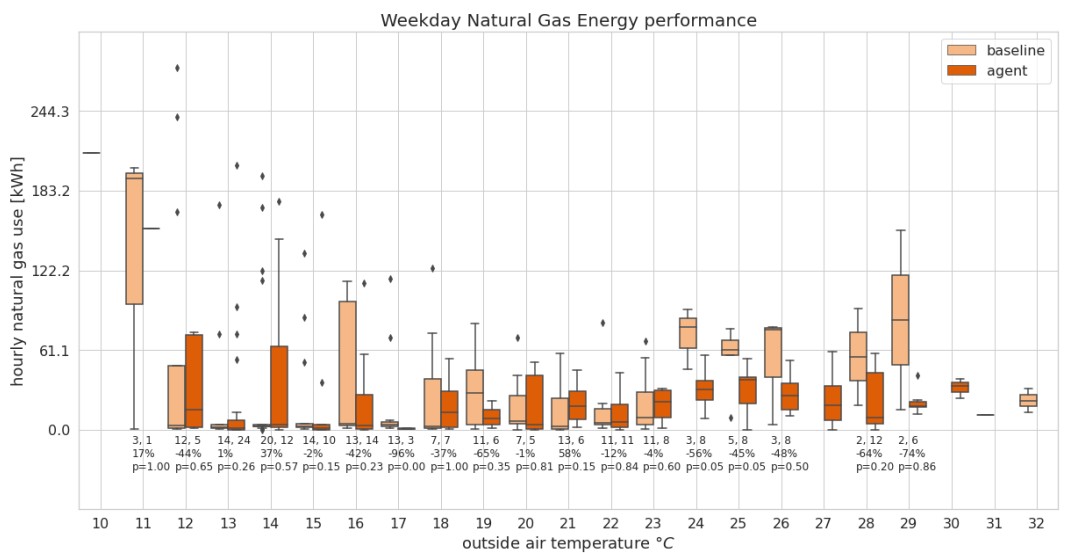

Figure 32: Natural Gas Energy Performance.

