# OpenReview forum: "Real-World Data and Calibrated Simulation Suite for Offline Training of Reinforcement Learning Agents to Optimize Energy and Emission in Buildings for Environmental Sustainability"
_ICLR.cc/2025/Conference — Submitted to ICLR 2025_

### Official Review · Reviewer_NPUo · 2024-11-01

**Soundness:** 3
**Presentation:** 3
**Contribution:** 3
**Rating:** 5
**Confidence:** 4

**Summary:**

The paper introduces the Smart Buildings Control Suite, an open-source suite for offline training of RL agents for energy and emission optimization in buildings. It provides real-world HVAC data from three buildings and a calibrated simulation environment compatible with OpenAI's Gym. The suite aims to bridge the gap between simulation and real-world application by providing scalable tools that emulate building dynamics and facilitate RL training. The authors demonstrate a novel calibration method and showcase baseline results using RL and time-series models.

**Strengths:**

- The availability of an open-source dataset and simulator addresses a critical gap in building energy optimization research by providing accessible benchmarks.
- The novel calibration method to align simulation with real-world data is a practical and relevant approach.
- The evaluation of predictive accuracy and its applicability for RL training adds to the suite's potential for broader adoption.
- The focus on optimizing building energy and emissions is timely and has significant implications for sustainable development.

**Weaknesses:**

- While the findings are validated through simulations, it would be helpful to understand if the authors have plans for real-world validation in future work. Could the authors discuss how the insights from their simulations might translate to real-world performance, possibly supported by prior studies or existing literature?
- The dataset, limited to three buildings in California, constrains the applicability of the results. It would be beneficial for the authors to include or plan for the inclusion of diverse building types or climates. Could they consider expanding their dataset in future work to encompass various climate zones or building structures, such as commercial or mixed-use buildings?
-  The focus on temperature calibration is clear, but it would be valuable for the authors to explain the rationale behind this choice. To enhance the study, could they incorporate or discuss additional energy consumption or emissions metrics, such as total energy use, CO2 emissions, or HVAC load efficiency?
- The lack of discussion on recalibration or retraining frequency is noted. Could the authors elaborate on potential strategies for adaptive recalibration, or provide an estimation of how often recalibration or retraining would be needed, based on their current experience with the data or anticipated building condition variations?
- A comparison with established methods like OCTOPUS [1] could provide valuable context for the proposed calibration method. Could the authors elaborate on specific aspects that differentiate or improve upon their method compared to the OCTOPUS approach or other similar frameworks, highlighting innovations or strengths?
- The benchmarking results provided are limited, with minimal comparative data between the baseline and the SAC-trained agent. Could the authors expand this section by including additional metrics such as total energy consumption, carbon emissions, or setpoint deviations? Visualizations like performance graphs or distribution plots would also enhance the understanding of the agent’s impact on overall building performance.
1. Ding, Xianzhong, Wan Du, and Alberto Cerpa. "OCTOPUS: Deep reinforcement learning for holistic smart building control." In Proceedings of the 6th ACM international conference on systems for energy-efficient buildings, cities, and transportation, pp. 326-335. 2019.

**Questions:**

- Could the authors provide more insights on how their simulator and RL agent would perform with different climates or building types?
- How robust is the simulator to noisy sensor data or partial information, which are common in real-world scenarios?
- Could the authors elaborate on the computational efficiency and practical deployment requirements of their suite?
- How adaptable is the calibration process to significant changes in building usage or occupancy patterns? Would recalibration require the same data volume and manual effort?
- Given that energy consumption and emissions are key objectives, why weren’t these included as primary metrics in the calibration?
- In the OCTOPUS paper[1], the calibration involved collecting detailed real-world data such as weather, occupancy schedules, and HVAC system parameters. Could the authors discuss how their calibration method compares to this approach?
- The Demonstration Benchmarking Results section reports limited metrics. Could the authors provide more comprehensive results including detailed metrics like energy consumption, carbon emissions, and setpoint deviations to better evaluate the policy's robustness and performance?

---

> ### Author Response · Authors · 2024-11-24
> **Response to Review**
>
> Thank you for your feedback! In response:
>
> 1. Yes, we do have real world plans! We allude to this briefly in the paper, but I am happy to elaborate more. We are going to live trials on this building in December, and if results are good, we will expand to more than just these 3. The reason why we think this will work in the real world is because we actually did have some small initial trials about a year back, but we have not had the ability to do so since.
>
> 2. We are already in the process of expanding to buildings in New York and London, and we hope to continually add more, and some of them are indeed mixed use! We view this publication as an inital way to get word out on this dataset, not as its conclusion. We think as it is, this is a worthy contribtion, but certainly more will be added, probably even some more in the next few months.
>
> 3. It certainly would be of value to include other metrics beyond temperature calibration. I am running right now calibration on C02 emissions, but the other suggestions will have to wait for future work.
>
> 4. Thank you for pointing out this lacuna. It was discussed, but very briefly, and clealy not sufficiently. We expect to recalibrate once per season in a temperate climate like California (4 times per year), based on our analysis of weather variance. In harsher climates, more than this may be needed, but that is not an analysis we have yet performed (hopefully when we have data on NY this can be addressed.
>
> 5. You asked for more comparisons with baseline, on energy consumption, carbon emissions, and setpoint deviations . All of this is already included in the paper. Perhaps you missed some of our charts?
>
> 6. Regarding the comparison to octopus, based on my analysis it is my understanding that the sensor data being used is not as granular and detailed.
>
> 7. Regarding the computational efficiency, our sim is super cheap, and can run a single iteration (5 min of sim time) in 2 seconds on a regular laptop.
>
> 8. Our sim is robust to some kinds of noise. We evaluated on scenarios where data is missing, but we did not test robustness on scenarios when the data is present but incorrect.

---

### Official Review · Reviewer_Vsqv · 2024-11-03

**Soundness:** 1
**Presentation:** 1
**Contribution:** 1
**Rating:** 3
**Confidence:** 4

**Summary:**

The paper proposes a dataset and simulation approach for building HVAC system. Reinforcement learning algorithms are implemented to test the efficacy of proposed approaches. While the proposed approach is technically sound and shows its capability on particular settings, the paper organization is relatively hard to follow, and the contributions in terms of constructing a new dataset or apply RL algorithms are not significant enough for the machine learning community.

**Strengths:**

* The paper collected six years of real-world historical data from three buildings, which is a valuable building data source.
* Proposed method considers the fine-grained ODE models.
* The proposed approach is claimed to strike a better balance between fidelity and execution speed.

**Weaknesses:**

* The paper does not properly address the challenge mentioned by the author, especially on the claim that "but training such a (RL) model, especially in a way that scales to thousands of buildings, presents many practical challenges." As the paper only shows limited device modeling and results on few buildings, it is very hard to justify this claim on scalability.
* The claim on "first open source interactive HVAC control dataset" is not rigorous. There are already several datasets and approaches working in this field.For instance, CityLearn, BEAR: Physics-Principled Building Environment for Control and Reinforcement Learning, and EnergyPlus all provided datasets.
* The main texts only show temperature comparison and a brief spatial errors on one building. These results cannot justify why the proposed method is more appropriate for HVAC RL tasks.
* A more thorough evaluation of modeling parameters, computation time, control performance, generalization to other buildings, comparison with state-of-the-art approaches is highly needed. For instance, compare with other building simulation software in terms of computation time and solution accuracy.
* The paper does not provide further insights nor algorithm development for HVAC control tasks. It is not clear why the proposed simulator is a proper fit for such tasks.
* The optimization of emission is considered in a very simple setting. A further analysis of carbon emission rate, different configurations of HVAC devices such as electric heat pump, water heaters or gas-fired heaters are needed.
* There is a need to demonstrate more detailed results on the efficiency of RL algorithms on these tasks.

**Questions:**

* How does the proposed framework compare to standard building simulation software such as EnergyPlus?
* How is the zone-to-zone heat interaction modeled in the paper?
* In Fig. 26, some predictions are really far off from historical data, can the authors comment on that?

---

> ### Author Response · Authors · 2024-11-24
> **Response to Review**
>
> Thank you for your comments. In response:
>
> 1. The claims of scalability stem from how rapid our simulator can be configured (3 hours per building) which is much faster than other competing sims.
>
> 2. None of the datasets you mentioned provide data at the level we do. Did you read the appendix that details how granular our data is?
>
> 3. While a more comprehensive evaluation would be nice, we believe as it stands, our work represents a novel dataset and benchmark that is missing from the community. In general, we beleive our work is an important contribution not from the perspective of the value of the agents we trained (which were mostly left to appendices), but due to the importance of our dataset and simulator as a new benchmark. Our simulator is easier to calibrate than existing work, lowering the bar for entry, and our dataset is far more detailed than any that exists, including the ones you mentioned. We ask you to evaluate our work from this perspective (ICLR in the CFP asks for datasets and benchmarks), and to look more closely and see that our dataset is indeed novel, even including the ones you mentioned (we mentioned many of them too in the paper!)
>
> 4. You wrote “The optimization of emission is considered in a very simple setting. ” This is a very surprising comment. It is certainly NOT a simple setting! It is a verty active area of research with dozens of papers published every year, and still no conclusive results. I respectfully disagree strongly with this comment.
>
> 5. Many of your questions are already addressed in the paper. We discuss energy plus at length in related works, and have a long appendix (appendix D) entirely dedicated to your second question.
>
> Thank you for your feedback, and if you feel we addressed your comments, especially since several were already discussed in the paper, we would appreciate if you would consider updating your score.

---

> ### Comment · Reviewer_Vsqv · 2024-11-25
> **Clarifications of Previous Reviews and Concerns about the paper**
>
> Thanks for your response. But I do still have concerns regarding the paper and the authors' response.
>
> C1. My concern was the paper claimed it could scale up to thousands of the buildings in the abstract. But the paper did not give a principled approach to show how the method can be scaled up. Moreover, in the main text, only a thermal model for water heater is shown. I doubt this can represent a universal building system.
>
> C2. I actually read the Appendix. Could you show in more details for the comparisons of data granularity? Because as far as the reviewer is aware of, both CityLearn and BEAR have 5-min resolution, while in EnergyPlus it can be configured for fine-grained simulation. At least the manuscript shall discuss the shortcomings of current off-the-shelf simulators. Moreover, Fig. 26 and Fig. 27 in Appendix are both blurred and without marking of x axis. Is that minute-level simulation or hour-level?
>
> C3. The reviewer is aware of the value of building dataset. However, the paper in current format does not provide enough novel insights or new simulation approaches. I would suggest the authors could set up common benchmarks and standard dataset creation methods, so that different building simulators and RL methods could be benchmarked in a more convenient way. So far the paper only shows simulation results for the author-defined environments.
>
> C4. Actually I am referring to the cost functions in Appendix A, and I suggest this optimization objective is very simple and far from reality. For instance, the grid emission level for electricity is changing with respect to both time and location. And if solar panels are considered for buildings, the emission level will be further complicated. The paper falls short of discussing all these perspectives.
>
> C5. I would suggest a comprehensive comparison with EnergyPlus or other off-the-shelf building simulators.

---

### Official Review · Reviewer_LCet · 2024-11-04

**Soundness:** 2
**Presentation:** 2
**Contribution:** 2
**Rating:** 3
**Confidence:** 5

**Summary:**

The authors provide a quick thermal diffusion simulation of a building and show that it can be calibrated with real data to provide a simulation that is somewhat close to real world data. The authors provide this as a jumping off point for people interested in training RL agents on building HVAC. The authors show that calibration reduces some error.

**Strengths:**

The framing of the problem is fair, and its an important problem to study. The notion of creating a fast and easily deployable RL training gym in building energy management is the correct path to study if RL is your goal.

**Weaknesses:**

- Quick note – “One of the most significant factors limiting progress is the lack of a reliable public benchmark to test solutions against.” In my experience, a far greater obstacle is just how far behind the hardware of building operating systems, live sensors, etc., are. Teams I know that have studied this wind up spending months just trying to get building sensors to talk correctly to RL agents, or have building systems accept actions from agents. I do not think you would believe just how many bugs there are in controls to hardware and how difficult it is to reliably detect anomalous building readings, be robust to flatlines, etc. So, acknowledging that even the perfectly trained RL agent would not be able to be dropped into a building, regardless of the quality / speed / versatility of the RL simulator, is more true to the practice of this field -- this detail is swept under the rug in this paper, which makes me believe that you are unaware of it. This should be included in the intro but also limitations and future works. And, you should eventually give it a shot if you'd like to see how hard it is!

- This RL gym is not referenced and seems to include more physics than your simulation. https://github.com/ugr-sail/sinergym.
Indeed, barely any of the works in Table 1: https://www.sciencedirect.com/science/article/pii/S2666546820300203 are mentioned in your background. I think you reference this paper when referring to EnergyPlus; they seem to do just fine in training an RL agent using EnergyPlus, so I’m not sure what scalability or configuration challenges you’re referring to. Have you tried these other approaches first before developing your own sim?

- You say that your simulation is physics inspired, but all I see is a thermal diffusion included via finite differences, and some building envelope information included. Fluid phase changes, solar radiance, and the effect of people / computers inside your building are all important physics to consider. These physics matter a lot! I don't see any discussion around how to include them (i.e. at the minimum, provide some thoughts on how to Monte Carlo the variation out could easily account for occupancy, for instance, and likely solar irradiance too. In practice, accounting for these additional physics scenarios doesn't even have to be a huge coding deal -- just some different scenarios that an MC setup can shuffle through.)  In a table, I would outline other major physics inputs and either say how they may be accounted for in future work, why they are unimportant, or what the trade-offs are in omitting them.

- When you visualize the spatial error, you don’t provide any statistical tests to back up significance. There are no error bars on your error either. Etc. Etc. Etc. There is no attempt to demonstrate anything beyond qualitative measures of success. Thus, the results are unscientific and hand-wavy. This is the biggest reason I chose a negative score. There isn't even much in the ways of a qualitative comparison to your uncalibrated simulation, or a baseline existing simulation, etc. How do we know your results are better?

**Questions:**

- You celebrate the fact that your n-step evaluation loss is novel, but to me, it reads like a very standard absolute value rollout error. Is the novelty the fact that you’ve summed over zones, and does that really deserve its own loss name? Furthermore, is summing over zones perhaps biasing your evaluation to zones that are smaller (seems like each zone is given equal weight in the sum)? Would normalizing by the size of the zone help?

- There’s no evidence (or discussion) from this work that agent transfer can happen with this dataset. Is this dataset is beneficial to anyone training an agent outside of these specific buildings; i.e. what is the value of this work to the community? Sorry, but I don't see this paper as relevant to the ICLR community. Buildings related conferences like ACM BuildSys (specifically, the Reinforcement Learning for Energy Management workshop) would be good fits for this work.

- how hard was it to instantiate a simulator with this specific building? I.e. how many engineer hours?

- Where is your code?

**Details Of Ethics Concerns:**

I do not believe that this is the case here, but in building data there can be sometimes concerns that personal privacy is compromised.

---

> ### Author Response · Authors · 2024-11-24
> **Response to Review**
>
> Thank you for your feedback. In response:
>
> 1. We are VERY AWARE of how hard this is! We have been in the process of setting this up since 2022! If we gave you any impression that we think running live on a building is a simple easy task, that was not intended. We have only gotten this to work recently with enormous hurdles. I do not disagree at all, but we are making the claim that we mitigate 1 major hurdle, not the only one! I am happy to rephrase our intro to make this clearer
>
> 2. Yes, we did explore most  of these options before trying our approach, and the issues always were:
>
> a. Too slow to configure to a large fleet of buildings
>
>  b. No easy way to directly integrate live data to the degree that we have it
>
> There are further details as well related to the specific needs of Google that these simulators were unable to meet, but I think that is also beyond the scope of this discussion
>
> 3. I agree with most of your comments pointing out the limits of our physical models. We DO account fot occupancy using a monte carlo simulations, perhaps you missed this section? But the other comments I agree with. However, I still feel this work is important and valuable in its current form. We are actively working on these improvements,
>
> 4. I understand your comments that our results are too hand wavy. However, repectfully I feel like this is missing the point of our work somewhat. We intended this to be a “Dataset and Benchmark” paper. We believe our important contribution that makes this work worthy of ICLR is:
>
>  a. A new and useful simulation that allows for easy access for people on a low compute budget to get into this space
>
>  b. An extremely detailed and granular dataset, the likes of which simply does not yet exist.
>
> The purpose of the results section, which was mostly left to appendices, was just to show the benchmark and dataset in action. However, the main contribution is certainly the data and benchmark itself. ICLR in the call for papers explicitly asks for new benchmarks, and that is the angle we are going for. I agree that on our results alone, more details are needed perhaps, but the tagline of this paper was not meant to be “Here are novel results on building control”, but rather “here is a new simulator to configure rapidly and lower the bar to entry, and a new dataset that is missing and very much needed”. I would ask that you evaluate our work from this perspective when providing a final score.
> However, I certainly want to improve, and all of your comments are worth doing. I was trying to get this all done by the rebuttal response deadline on the 26, but it will take me at least another week to add all your suggestions to the draft to make this more precise. However, kindly evaluate this as a dataset paper, and I think that our work meets the standards of quality.
>
> 5. Yes, the dataset is certainly beneficial outside to people working on this buildings!  Since release, over 70 people have reached out to me via linkedin, email, and in person, asking me how they can use this data. I genuinely do not understand why you think this data is not of use, and I think perhaps this disconnect is why you did not evaluate our paper as a datasets and benchmarks work. In about two months our simulator has gotten 50 stars on github (cannot share link due to double blind), ML researchers at 11 universities have reached out to us about it, and the NSF is interested in funding a competition. There is enormous community interest, and frankly it is not clear to me why you think this would not be the case.
>
> 6, It takes 3 engineer hours for each building.
>
> 7. The code is missing because we were not able to anonymize it. However, it is on github and is already being used by several universities.

---

### Official Review · Reviewer_3j5q · 2024-11-05

**Soundness:** 2
**Presentation:** 3
**Contribution:** 3
**Rating:** 5
**Confidence:** 4

**Summary:**

The work presents The Smart Buildings Control Suite, a benchmarking tool for the optimal control problem of HVAC devices for reducing energy consumption and environmental impact. The contribution is composed of two parts: a real-world historical HVAC dataset relative to three buildings and a highly customizable and scalable low-to-medium fidelity HVAC simulator. Authors employ the aforementioned tool, calibrated by the usage of real-world historical data, in three different ways:
- training an RL agent and evaluating it against a baseline
- learning a dynamic model through an LSTM encoder-decoder network
- training an RL agent on the previously learned dynamic model.

**Strengths:**

- Concept: HVAC consumption is a hot research topic and authors have clearly highlighted the problem and described the contributions of their work.
- Literature: the authors have presented an exhaustive review of works related to the topic.
- Presentation: the paper is clear and well-written, and the authors provide a very detailed description of every aspect of the framework.
- Collected data: the authors dispose of many real-world data (six years) from three different buildings which contribute to calibrating the simulator.
- Simulator: low-to-medium fidelity physics-based simulator, thought to be lightweight to foster RL experiments.

**Weaknesses:**

- The RL agent is compared to a **control baseline which is never specified in the paper**. Understanding the policy of the baseline is essential to evaluate the RL method and authors may provide a detailed description of the baseline control strategy and its decision-making logic.
- Apart from the unspecified baseline, **the RL agent is not compared with other RL and non-RL strategies**. For example, even simple rule-based policies, such as a fixed action policy, can be considered as additional baselines and can prove the need for an RL solution for the tackled problem. Regarding RL methods, I could suggest trying PPO, A2C, and DDPG, for example.
- The simulator has been calibrated over two days of historical data and validated on a third one for building SB1. While this may be enough to capture building behavior across a day-night cycle, it surely **cannot be sufficient to deal with the building's yearly behavior**, which is affected by seasonal weather and the alternation of work days and weekends. Authors may discuss more in detail how they plan to address seasonal variations and long-term patterns.
- **The training of the RL agent is not completely clear.** From what the authors write, it seems that the agent is trained with the trajectories generated by the simulator calibrated on two days of data (thus almost 580 samples) and trained over 4000 samples, which seems too few for a deep RL agent to learn an optimal policy. I would suggest providing further details about the training process, including how the agent has been fed with simulated time series, how many samples have been used, and how train and test series have been split.
- Since each building behaves differently due to its structure, the **simulator has to be calibrated for each specific building and cannot be generalized**.
- Some **plots are not very clear**. In particular, figure 26 does not have an explicit legend, figure 27 is pixelated and labels on the axes are not readable, and figure 28 is pretty difficult to understand.
- Supplementary materials containing the **codebase or scripts have not been provided**.

**Questions:**

- I did not understand why the discount factor $\gamma$ is considered in the hyperparameter tuning phase of SAC, since $\gamma$ is not a hyperparameter of the problem, but rather a part of the MDP that defines the problem, thus definitely not something to tune.
- I would ask you to explain figure 18, since I don't understand if the lines represent the cumulative reward of the test episode or the total return across the training phase. Moreover, I am not convinced by the fact that the baseline line is perfectly horizontal. Could you provide an insight about that?
- I noticed that several simulator's hyperparameters are located on the edges of their interval range. Is it something you find reasonable? Are you fine with this outcome or could it lead to undesirable behaviors?
- Based on Figure 26, you stated in Appendix I, lines 2432-2433, that the LSTM model "provides good correspondence with the actual real data signals". However, some signals in these plots are significantly mismatched. Could you provide some numerical metrics and confidence intervals to corroborate your statement?
- I'm not convinced by Figure 20: the $y$ axis has all ticks set to $0.0$ and I cannot understand how to properly read the plot. Could you maybe add a line plot with confidence intervals to compare the baseline and SAC agent, highlighting the horizontal range considered "comfort temperature"? This would be nice to have also for the last experiment of SAC trained with the LSTM model, to confirm that the SAC policy maintains the comfort temperature by reducing carbon emissions, electrical energy, and natural gas energy consumption.
- I did not completely understand the sentence in lines 519-520: "we also trained an RL agent on the learned dynamics model, demonstrating the ability to learn a policy directly from data without involving the simulator". The LSTM model is a simulator, a data-driven one. Moreover, RL is based on training from data via a trial and error approach, thus there is no need for you to demonstrate this.

---

> ### Author Response · Authors · 2024-11-24
> **Response to Review**
>
> Thank you for your feedback. In response:
>
> 1. We believed that baseline control policies in commercial buildings today are well described in the literature. However, in light of your feedback, perhaps this was an oversight. If accepted, we will include this in the final draft.
> 2. Since submission, we have tried PPO, SAC, rule based policies, and model based Monte Carlo Tree Search. For the final version, we can add a table with all these comparisons. All performed better than the baeline (which we now have a description of, as mentioned above.
> 3. You are absoultely correct that 2 days is not sufficent. To address this, we updated our experiments and made the following changes:
> We ran experiments on 2 weeks of training and two of validation, for one month total. The results still support our main thesis
> We added a section outlining the strategy for how we intend to scale this yearly. In short, we intend to train a different model for each climate. Meaning, one for summer, one for winter, etc. We think based on our initial experience this will perform better than a single, all season, model.
> 4. Thank you for pointing out the missing training details. We have expanded this section. We calibrated on one month of data, as mentioned above, and trained now on 2 weeks of simulated data. All of the details are now included. The code is also public
>
> 5. You mentioned that the simulator cannot be generalized to other buildings. Respectfully, we feel this is not a weakness of our method, but rather one of the central challenges of this entire field. We do not claim our simulator is the complete solution, just a strong step in the right direction. Our simulator is easier to calibrate and configure to a new building than most, but indeed such steps still need to be taken. We are thinking very carefully about how to generalize to new buildings, but as of yet no one has been able to solve this issue.
>
> 6. I have clarified the figures and added longer descriptions
>
> 7. Regarding the code base: This is a Google open source project. The code is already available online, and has about 50 stars on github. We simply could not provide it without breaking the double blind review process. However, it is already public and being used
>
> In response to your questions:
> 1. Perhaps this should not have been a hyper parameter. However, I will explain the logic. Unlike in some other RL paradigms, this is a multi objective reward, cost carbon and comfort. As such, it is very non trivial how to weigh the reward function in such a fashion as to actually balance these objectives. Different reward decays need to be tested and evaluated to ensure all the objectives are being met, in addition to careful weighing of all the reward components.
> 2. Figure 18 shows the total reward earned by an agent over the evaluation episode. The baseline does not change, so it earns a constant performance on the evaluation episode, since it is the same each time. What perhaps confused you on this chart is that some methods seem to perform better in the beginning, and then actually get slightly worse over time. I think this is due to a to the agents overfitting the train episodes, and thus doing worse on the eval episodes. To fix this, we are currently re running the RL on a much longer train window ( two months of simulated time.). I will have this done within 2 weeks, before the camera ready deadline
> 3. Regarding hyper parameters being located on the edge of intervals: This is a fantastic question. We considered expanding the intervals, but this leads to the issue of, do we prioritize performance, or physical plausiblity? We beleive that the model wants to make some paramters have values that are not plausible, to compensate for other modeling wekanesses. However, allowing for expanded intervals is not just physically implausible, but led to worse generalization. Still, we agree this needs further investigation.
> 4. You are correct. Figure 20 axis was not labeled properly. My apologies. I will adjust the figure.
> 5. I am not sure what part of that sentence was unclear, but I will try and explain. The LSTM is certainly also a simulator. I simply meant that it is a simulator fully reliant on data, and not at all physics based, and I wanted to demonstrate that our agent learned across varied environment types.

---

> > ### Comment · Reviewer_3j5q · 2024-11-26
> >
> > Dear authors,
> > thank you for addressing my questions and concerns.
> >
> > Unfortunately, I will still leave my rating unchanged - marginally below the acceptance threshold.
> >
> > While I believe your work is valuable, there is still room for improvement. Specifically, greater emphasis should be placed on the RL experimental campaign. It seems that RL has been treated somewhat as a black-box tool, whereas in real-world settings, explainability often makes a critical difference. Lastly, I expected to be provided with a revised version of the paper during the rebuttal phase. Without this, I am unable to evaluate the changes you have made effectively.

---

### Meta-Review · Area_Chair_7xF3 · 2024-12-21

**Metareview:**

This paper presents the Smart Buildings Control Suite, a benchmarking tool that combines a real-world HVAC dataset and a configurable simulator for optimizing building energy consumption through reinforcement learning. While the work addresses an important practical challenge, it has significant limitations including inadequate experimental validation, insufficient comparison with existing solutions, and unclear differentiation from prior simulation approaches.

**Additional Comments On Reviewer Discussion:**

During the discussion, reviewers raised several key concerns: (1) insufficient validation of the simulator's capabilities and comparison with existing solutions like EnergyPlus and CityLearn, (2) limited scope of the physical models which omit important factors like fluid phase changes and solar radiance, and (3) lack of rigorous statistical analysis in the results. The authors responded by highlighting their simulator's rapid configurability and dataset granularity, but did not adequately address all the concerns of the reviewers.

---

### Decision · Program_Chairs · 2025-01-22

Reject